# Cryo-EM structures of higher order Gephyrin oligomers reveal principles of inhibitory postsynaptic scaffold organization

Diego Ortiz-López [1], Tamsanqa T. Hove [1], Christiane Huhn [1], Serena Camuso [2], Pia M. van gen Hassend [1], Bodo Sander[1,5], Benjamin F. N. Campbell [3], Shiva K. Tyagarajan [3,6], Andreas Plückthun [4], Christian G. Specht [2], Hans M. Maric [1], Bettina Böttcher [1] & Hermann Schindelin [1] ✉

Gephyrin, the principal scaffolding protein of inhibitory postsynaptic densities, clusters glycine and GABA$_A$ receptors via multivalent interactions. It features structured N and C terminal domains connected by an intrinsically disordered linker. Although the structural and functional properties of its terminal domains are well characterized, the mechanism by which full-length gephyrin organizes into higher-order complexes remains unresolved. Here, we combine biochemical reconstitution, cryo-electron microscopy, and mutational analyses to elucidate the structural logic of gephyrin oligomerization. We demonstrate that gephyrin adopts a stable dimeric assembly which constitutes the basic unit for both linear and oblique tetramers as well as linear hexameric arrangements. High resolution structures reveal a critical segment of the flexible linker that adopts two distinct conformations, one of which occludes the receptor-binding site. This segment harbors key phosphorylation sites, suggesting a regulatory control mechanism. Our findings redefine the architecture of inhibitory postsynaptic sites and reconcile gephyrin oligomerization models with published in-situ postsynaptic densities characterized by cryo-electron tomography.

Inhibitory neurotransmission plays an essential role in balancing excitatory signals and ensuring proper circuit functioning in the central nervous system. Fast synaptic transmission relies on a finely tuned interplay between presynaptic neurotransmitter release sites and cognate neurotransmitter receptors, which are enriched by scaffolding proteins at postsynaptic sites. At inhibitory synapses, gephyrin is the main scaffolding protein[1-4], determining the clustering and positioning of both glycine and γ-aminobutyric acid type A receptors (GlyRs and GABA$_A$Rs) by direct binding[5-12]. Gephyrin is a soluble protein of ~80 kDa (Supplementary Fig. 1) composed of a 20 kDa N-terminal domain (GephG), a central unstructured linker and a 45 kDa C-terminal domain (GephE)[13]. The existence of multiple splice variants, affecting primarily the linker region but also the structured domains, gives rise to substantial protein heterogeneity[14].

Gephyrin interacts with GlyRs, specifically the β-subunit, and with the GABA$_A$R α1-3 subunits through GephE, which binds to the

[1]University of Würzburg, Rudolf Virchow Center for Integrative and Translational Bioimaging, Josef-Schneider-Str. 2, Würzburg, Germany. [2]Neuro-Bicêtre, Inserm U1195, Université Paris-Saclay, Bâtiment Gregory Pincus, 80 Rue du Général Leclerc, Le Kremlin-Bicêtre, France. [3]University of Zürich, Institute of Pharmacology and Toxicology, Winterthurerstr. 190, Zürich, Switzerland. [4]University of Zürich, Department of Biochemistry, Winterthurerstr. 190, Zürich, Switzerland. [5]Present address: CSL Innovation GmbH, Marburg, Germany. [6]Present address: Department of Life Sciences, Center for Neuroscience and Cell Biology, University of Coimbra, Coimbra, Portugal. ✉e-mail: hermann.schindelin@uni-wuerzburg.de

intracellular loop located between transmembrane helices 3 and 4 of the respective receptor subunits. Structurally, GephE is composed of four subdomains (I-IV; see Supplementary Fig. 1)[15–17] with the receptor-binding site being located at the interface between subdomains III and IV of one monomer and subdomain IV of the opposing monomer[16,17]. In addition to receptor binding, gephyrin also engages with the cytoskeleton, directly by binding to microtubules[18], and, indirectly, through adapter proteins. These interactions are mediated by dynein light chains and members of the Ena/VASP family, which regulate actin filament dynamics[19,20]. Furthermore, gephyrin catalyzes the final two steps of molybdenum cofactor (Moco) biosynthesis: the adenylation of molybdopterin and the subsequent incorporation of molybdenum to form the mature cofactor[21].

Liquid-liquid phase separation (LLPS) has been described as a mechanism through which gephyrin may control the dynamic organization of synaptic components[22,23]. Through multivalent interactions, gephyrin can form condensates that facilitate receptor clustering, thus potentially regulating inhibitory synaptic plasticity. It has been proposed that direct interactions between the linker and subdomain II of GephE modulate gephyrin LLPS in the presence of artificially dimerized loops derived from the GlyR β-subunit[22]. Detailed biochemical studies or structural data pinpointing the molecular interactions of linker regions and subdomain II have been largely missing until now. Lee et al. [22]. demonstrated a critical role of gephyrin linker residues 286 to 326 for gephyrin condensate formation in HEK293T cells as well as inhibitory synapse targeting and function in primary neurons. The complete absence of structural information about the linker, however, is a limitation in confirming the mechanism of gephyrin scaffolding at synaptic sites.

Full-length gephyrin (GephFL) has been proposed to form a trimer, mediated by GephG[24,25], whereas the dimerization of GephE, consistently observed for GephE in isolation, was thought to be suppressed by an unknown mechanism in the context of the full-length protein[17]. For a long time, this trimeric model served as the foundation for proposed architectures of the inhibitory postsynaptic density (PSD), including both hexagonal lattice[15] and irregular network[17] models. More recently, however, a semi-ordered gephyrin meshwork has been suggested based on cryo-electron tomography (CryoET) reconstructions of GABA$_A$ receptors at hippocampal inhibitory synapses[26]. This newer model posits dynamic transitions between gephyrin dimers and trimers, as trimer-only configurations fail to account for the full range of observed interreceptor distances, angles, and spatial distributions.

In this study, we provide biochemical and structural evidence that GephFL, when purified following expression in *Escherichia coli*, can adopt a stable dimeric state. In addition, we isolated higher-order oligomers, specifically dimers of GephE-mediated dimers. We further determined cryo-EM structures of (1) isolated GephE and (2) GephE within GephFL in complex with a DARPin named 27F3[27], and (3) higher-order assemblies consisting of GephE-mediated dimers of dimers. The structure of GephFL revealed how C-terminal residues of the linker interact with subdomains II and III of GephE, while the dimer of dimers structure elucidates the architecture of higher-order gephyrin oligomers. Site directed mutagenesis of residues at key interaction sites identified in these structures demonstrated that both the intermolecular dimer-dimer interface and the intramolecular GephE-linker interface are essential for gephyrin clustering in HEK293T cells and hippocampal neurons. Together, these findings uncover previously unknown modes of gephyrin oligomerization that explain the organization of the inhibitory postsynaptic density at GABAergic synapses as observed in electron cryo-tomography reconstructions and define molecular interfaces that underlie these assemblies[22,23].

## Results

### Full-length gephyrin exists in different oligomeric states in solution

Based on the published purification protocol for the P2 splice variant of gephyrin[28], and considering the known heterogeneity of gephyrin preparations[28], we developed a modified protocol (see "Methods" and Fig. 1a) to obtain homogenous samples of the P1 splice variant suitable for single particle cryo-EM analysis. Following anion exchange chromatography, we consistently observed two overlapping peaks (fractions A and B) and a trailing shoulder (Fig. 1b). These fractions were subjected separately to a first round of size exclusion chromatography (SEC; Fig. 1c). Fraction A eluted as a leading shoulder followed by three overlapping peaks (Fig. 1c, blue trace). Among these, fraction A4 was excluded from further analysis as it did not run as a single peak in the size exclusion chromatography step (Supplementary Fig. 2). Fraction B produced a major peak (designated B1) along with a trailing shoulder (Fig. 1c, black trace). Since fraction A3 eluted at a similar volume as fraction B (Fig. 1c), it was not analyzed further.

Blue native PAGE (BN-PAGE) analysis of fractions A and B (Fig. 1d) revealed that fraction A was enriched in higher-order oligomeric states. Following the first SEC step, sample quality was notably improved. BN-PAGE analysis of fractions A1, A2 and B1 (Fig. 1d) showed that fraction B1 contained a prominent band migrating between the 146 kDa and 242 kDa markers, the dominant species of fraction A2 migrated between 242 kDa and 480 kDa and fraction A1 was enriched in species migrating above 480 kDa. A second round of SEC on fractions B1 and A2 further improved sample purity, as indicated by single, well-defined peaks (Fig. 1e), resulting in refined fractions B1′ and A2′. BN-PAGE analysis (3–12%) of fraction B1′ showed a single band migrating between the 146 kDa and 242 kDa markers (Fig. 1f).

Fractions B1′, A2′, and A1 were subsequently analyzed by SEC-MALS using a Superose 6 column. These analyses yielded molecular masses of 168 kDa for fraction B1′ (Fig. 1g), 350 kDa for fraction A2′ (Fig. 1h), and 480 kDa for fraction A1 (Fig. 1i). The molecular mass of B1′ aligns well with the expected mass of a GephFL dimer (~170 kDa). The mass readings for A2′ and A1 were less consistent across their respective peaks (Fig. 1h, i), which was at least in part due to the presence of a strongly scattering high molecular mass peak (Supplementary Fig. 2). Nevertheless, the data suggest that these fractions correspond to higher-order oligomers approximately twice (A2′) and three times (A1) the mass of a GephFL dimer. Notably, the properties of fraction B1′ suggested that a GephFL dimer represents the basic building block for the formation of higher-order gephyrin oligomers, including tetramers and hexamers. It should be pointed out that the 168 kDa mass obtained for the P1 variant is in excellent agreement with recent SEC-MALS data from Bai et al. [22], who reported a value of 172 kDa. Earlier mass estimates from our lab and others, which concluded that gephyrin primarily forms trimers, may reflect the use of the P2 splice variant, the presence of mixed oligomeric species resulting from differing purification protocols, and/or transitions between trimers and dimers. The latter would be in agreement with two main bands between 242 and 480 kDa observed in Fig. 1d, as expected molecular weights for the trimers and dimer of dimers are ~250 kDa and 340 kDa, respectively.

### DARPin 27F3 specifically binds both GephFL and GephE

Designed ankyrin repeat proteins (DARPins) are engineered binding proteins known for their high stability, modular architecture, and strong affinity for their targets, thus rendering them valuable tools in structural biology[29]. Recently, several gephyrin-specific DARPins were developed, including 27F3, which binds GephE with high affinity[27]. Given their repetitive structure and well-defined shape, DARPins can also serve as molecular fiducials in cryo-EM, where their presence improves particle alignment and aids in structure determination, particularly for small or flexible proteins[30–33]. To assess the suitability of 27F3 for facilitating cryo-EM structure determination of gephyrin,

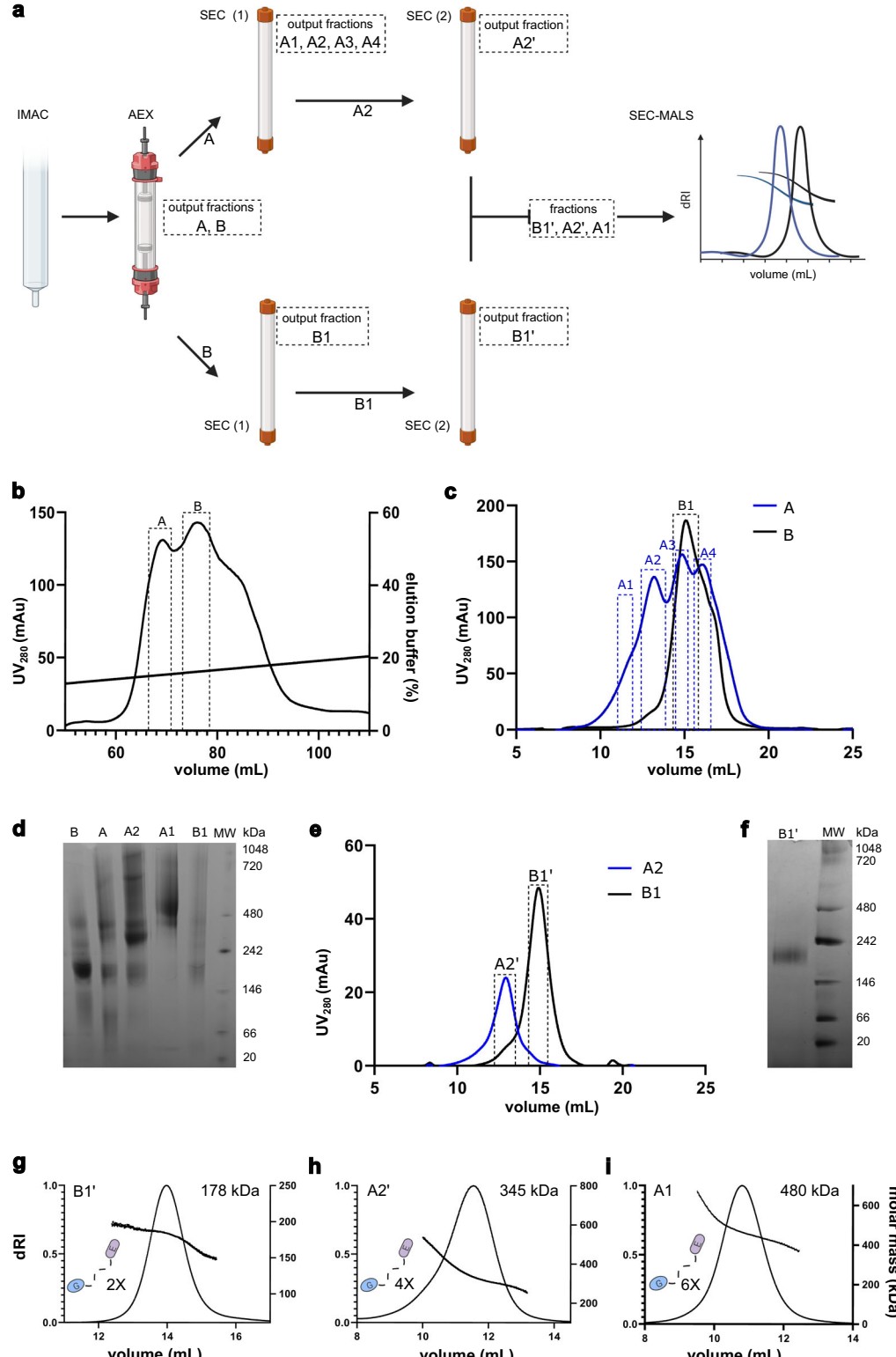

**Fig. 1 | Gephyrin adopts different oligomeric states. a** Scheme of the purification protocol with the resulting fractions after each step indicated. This panel was partially created in Biorender; Ortiz-Lopez, D. https://BioRender.com/464mroy. **b** Mono Q anion exchange chromatography yielded fractions A and B. **c** Analysis of fractions A and B by size exclusion chromatography (SEC) revealed further heterogeneity resulting in fractions A1-A4 and B1. **d** Blue native (BN) PAGE analysis (3–12% gradient gel) of the fractions obtained after anion exchange (fractions A and B) and selected fractions after the first SEC (fractions A1, A2 and B1).

**e** Chromatogram of fractions A2′ and B1′ after the second SEC. SDS-PAGE analysis of these fractions is shown in Supplementary Fig. 2. **f** Analysis of fraction B1′ by BN-Page (3–12% gradient gel). **g–i** SEC-MALS analyses of fractions B1′ (**g**), A2′ (**h**) and A1 (**i**). Molecular weights are indicated, and they are compatible with two (x 2), four (x 4) and six (x 6) times the mass of a gephyrin monomer, as indicated on the left of each chromatogram. Panels (**g–i**) were partially created in Biorender; Ortiz-Lopez, D. https://BioRender.com/pus7ffd. The purification process was repeated 3 times, obtaining similar results. Source data are provided as a Source Data file.

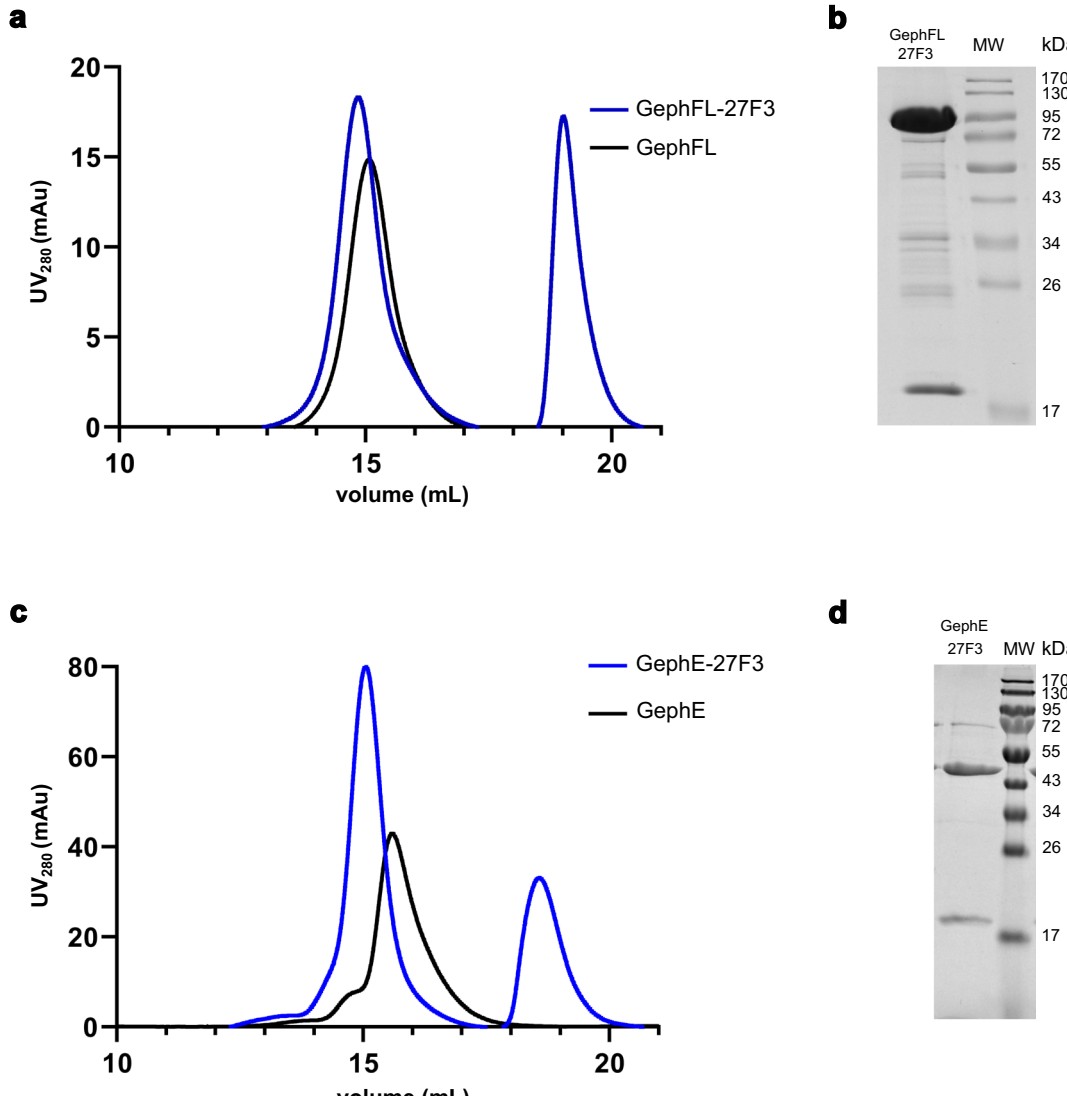

**Fig. 2 | DARPin 27F3 interacts with full-length gephyrin and GephE. a** Size exclusion chromatography of GephFL in isolation (black) and in complex with 27F3 (blue). **b** SDS-PAGE analysis (15% gel) of the corresponding peak fractions. GephFL migrates at 82 kDa, 27F3 at 19 kDa, intervening bands represent degradation products of GephFL. **c** SEC of GephE in isolation (black) and in complex with 27F3 (blue). **d** SDS-PAGE analysis of peak fractions (GephE migrates at 47 kDa). SEC experiments were repeated three times, obtaining similar results, individual representative chromatograms are shown in both the (**a** and **c**) panels. Source data are provided as a Source Data file.

we analyzed its interaction with both GephFL and GephE using SEC. In both cases, complex formation with 27F3 resulted in a clear shift toward higher molecular weight species (Fig. 2a, b). SDS-PAGE analysis of the corresponding SEC fractions confirmed the presence of both 27F3 and gephyrin in the shifted peaks, thus indicating stable complex formation (Fig. 2c, d). Consistent with previous reports[34], GephFL exhibited signs of proteolytic degradation, likely due to the exposed and flexible linker region, which is known to be particularly sensitive to protease activity.

### Cryo-EM structure of the $GephE_{309}$-27F3 complex

Encouraged by the high binding affinity between 27F3 and GephE, we next sought to determine the structure of the $GephE_{309}$-27F3 complex using cryo-electron microscopy (cryo-EM), as described in Supplementary Fig. 3. The resulting density map was resolved to a final resolution of 2.3 Å and the structure was refined by real-space refinement in Phenix (Supplementary Table 2), using the crystal structure of GephE (PBD entry 4PD0) and an AlphaFold-generated model of DARPin 27F3 as starting models.

In the complex, one 27F3 molecule binds asymmetrically to a GephE dimer, simultaneously engaging subdomain IV from both monomers (Fig. 3a). Only the internal repeats 1–3 and the C-capping repeat of the DARPin are resolved in the cryo-EM map, suggesting that the N-terminal capping repeat is disordered. The primary binding contributions come from repeats 1, 2, and 3. The complex buries an interface of 1077 Å², corresponding to 16% of the accessible surface area of 27F3 and 5.2% of $GephE_{309}$, with the majority of the buried surface contributed by the A-chain of $GephE_{309}$. The intermolecular interactions are dominated by hydrophobic contacts between $GephE_{309}$ residues P685, L686, P713, P714 and V727, and 27F3 residues V47, V48, and the methylene groups of K147 (Fig. 3b, c). These hydrophobic contacts are complemented by polar interactions involving $GephE_{309}$ residues H682, Q683, R660, and D729, and 27F3 residues D46, N83, W58, Y91, and R114 (Fig. 3d).

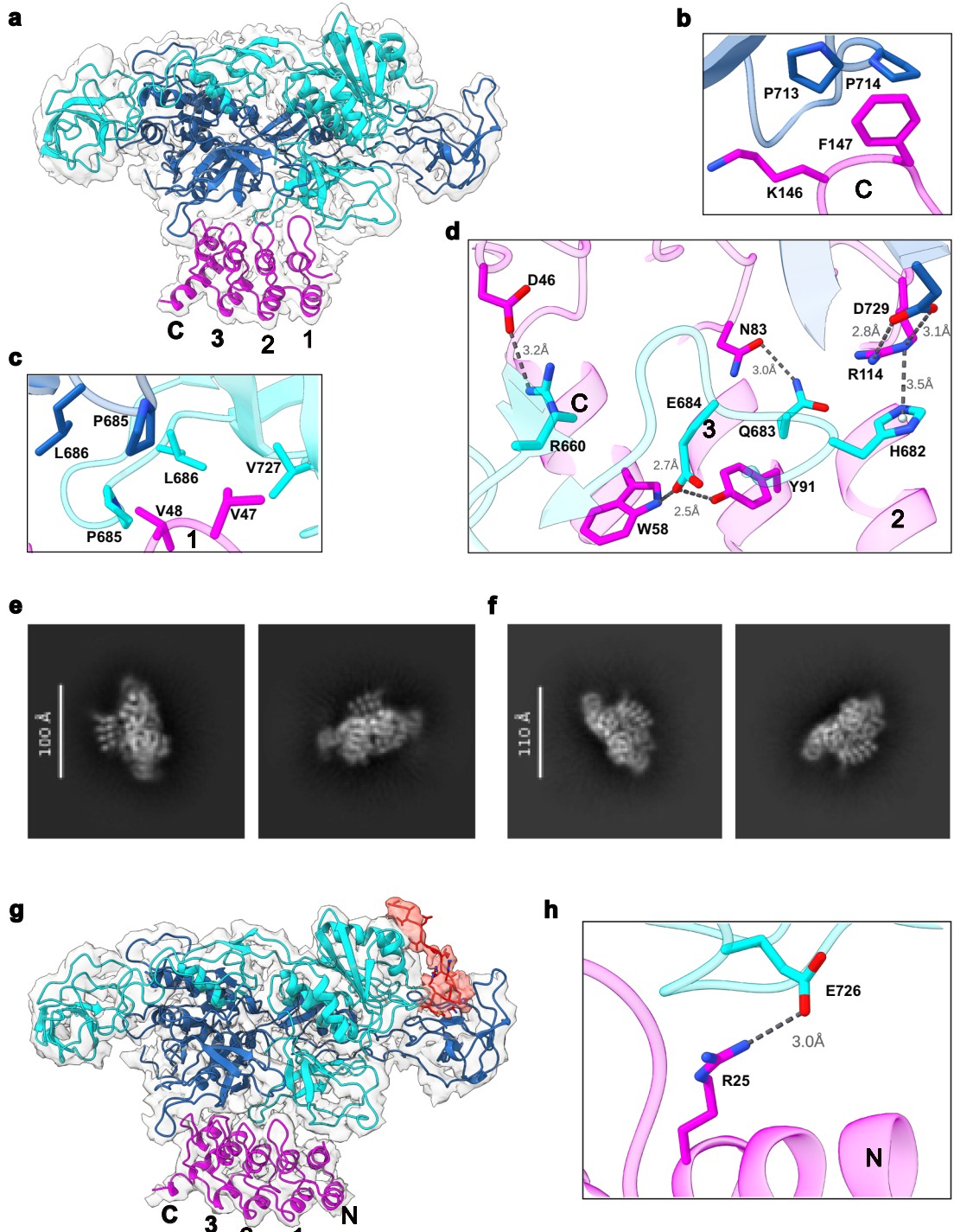

**Fig. 3 | Structures of GephE_{309}-27F3 complexes. a** Overall structure of the GephE_{309}-27F3 complex in ribbon representation together with the EM-derived volume. The two subunits of GephE are shown in blue and cyan, respectively, and 27F3 in magenta with the internal repeats labeled 1–3 and the C-capping repeat with C. **b–d** Enlarged view of crucial van der Waals (**b**, **c**) and polar (**d**) interactions in the interface with important residues highlighted in stick representation. **e**, **f** 2D classes revealing either the presence of four (**e**) or five (**f**) ankyrin repeats in the GephFL-27F3 complex. **g** Structure of the GephFL27F3 complex, where gephyrin was present as the full-length protein, thus allowing visualization of parts of the linker. Ribbon representation color-coded as in (**a**) with residues 305–319 in the linker and its corresponding density in red. The N-capping repeat of 27F3, which is visible in this structure, is labeled with N. **h** Interface of the complex involving residues in the N-capping repeat.

## Cryo-EM structure of the GephFL-27F3 complex

To assess whether binding of 27F3 to GephFL is influenced by additional residues in the linker, and to determine whether direct GephE-linker interactions could be visualized, we determined a cryo-EM structure of the GephFL-27F3 complex. Although the full-length protein was used, we could not deduce a high-resolution structure for GephG. However, high-quality 2D classes were obtained for the interaction between GephE (in the context of GephFL) and 27F3. Notably,

two distinct sets of 2D classes emerged, showing either four or five resolved ankyrin repeats in 27F3 (Fig. 3e, f). Further analysis using CryoSPARC, including heterogeneous refinement and 3D variability analysis (Supplementary Fig. 4), revealed a subpopulation of particles with additional densities near subdomain II of GephE (Fig. 3g), which will be described in the next section.

Model building and refinement in the resulting 3.4 Å map was initiated as described for the GephE$_{309}$-27F3 complex. The map revealed density for the N-capping repeat as anticipated from the 2D classes. An analysis of the B-factors resulting from real-space refinement in Phenix showed that the average B-factors of the atoms in the N-capping repeat were considerably higher than those of the atoms in the remaining repeats, reflecting that the N-capping repeat can be either ordered or disordered, as demonstrated by the 2D classes and structures of the GephFL/E-27F3 complexes. Structurally, only a single salt bridge was observed between the N-capping repeat and GephE, involving R25 in 27F3 and E726 in subdomain IV of GephE (Fig. 3h). The limited interaction is not unexpected, as the N- and C-terminal capping repeats of DARPins are constant across designs[29], while the randomized regions are located in the internal repeats (positions 12–35), which mediate the majority of specific target interactions[27]. As a result, nearly all binding contacts between 27F3 and GephE involve internal repeats 1–3. Despite the presence of full-length gephyrin, the GephE-27F3 and GephFL-27F3 complexes are nearly identical and can be superimposed with a root mean square deviation (rmsd) of 0.74 Å. This indicates that the additional residues in GephFL do not alter the GephE-27F3 interaction interface.

Given the proximity of 27F3 to subdomain IV, an essential region for receptor binding, we assessed the compatibility of 27F3 binding with receptor engagement. To do this, we superimposed the GephFL-27F3 complex onto the structure of GephE bound to a GlyR β-subunit-derived peptide (PDB entry 4DP1), where two peptides bind symmetrically to the GephE dimer. The superposition revealed that 27F3 sterically occludes the receptor binding site on the B-chain, while the site on the A-chain remains accessible (Supplementary Fig. 5). This suggests that 27F3 may partially interfere with receptor binding and may also influence GephE oligomerization with other binding partners by bridging the two subunits. In summary, the cryo-EM data demonstrate that GephFL forms a stable complex with 27F3 through its E-domain, with no apparent structural influence from either the linker region or GephG. The structural core remains consistent with the isolated GephE$_{309}$-27F3 complex, although the increased flexibility in full-length gephyrin likely limits resolution of the entire architecture. Importantly, this structure provides direct evidence of GephE-linker interactions in full-length gephyrin, as discussed in the following section.

### The C-terminal region of the linker interacts with and stabilizes GephE

The cryo-EM structure of the GephFL-27F3 complex revealed previously unresolved features at the interface between the linker region and GephE. Specifically, we observed additional density corresponding to residues 305–319, representing the C-terminal portion of the linker (Fig. 3g). This finding prompted a detailed investigation into the structural and functional contributions of this region. This extra density was present in both monomers of the GephE dimer; however, the map quality was significantly better for the subunit referred to as the B-chain, and therefore, model building was performed only for this monomer. The resolved linker segment wraps around subdomain III of monomer A (Fig. 4a). The first visible residue (S305) is located near residues N552, L555 and N556. The hydrophobic residues V307 and I309 engage in van der Waals interactions with P551, L555, and V585 in subdomain III (Fig. 4b). This hydrophobic stretch is followed by a series of positively charged residues, R314, R315, H316, and R317, which form electrostatic interactions with

residues E442 and E578 in subdomain II of the B-chain (Fig. 4c), i.e., the other monomer. Notably, the electron density for the hydrophobic region is better defined than that for the charged segment, possibly reflecting flexibility or variability in the electrostatic interactions.

To assess whether these linker residues stabilize the GephE domain, we performed unfolding experiments using differential scanning calorimetry (DSC) and circular dichroism (CD) spectroscopy on two constructs: GephE$_{318}$ (residues 318–736) and the N-terminally extended variant GephE$_{309}$ (residues 309–736) (Fig. 4d, e). GephE$_{318}$ unfolded at 56.9 °C (DSC) and 58.1 °C (CD), whereas GephE$_{309}$ showed an elevated melting temperature of 60.5 °C in both assays. This corresponds to an increase in thermal stability of 2.4-3.6 °C, indicating that the linker region has a stabilizing effect on GephE. As shown in Supplementary Fig. 6a, there are no significant differences between the CD spectra of GephE$_{318}$ and GephE$_{309}$, which agree with a protein containing a mixture of α-helices, β-sheets and loop elements. In addition, a comparison of representative SEC chromatograms for both variants (Supplementary Fig. 6b) indicates no evidence of higher-order oligomerization.

To further probe GephE-linker interactions, we conducted peptide microarray experiments (Fig. 4f). These confirmed that the basic motif $^{314}$RRHR$^{317}$ in the linker is essential for interaction with the negatively charged subdomain II of GephE. Peptide fragments lacking this motif displayed substantially reduced binding, highlighting a critical role of these residues in mediating electrostatic interactions. Additional regions within the second half of the linker also showed GephE binding on the array and contain positively charged residues. However, since these segments were not resolved in the cryo-EM structure, they will not be discussed here.

In summary, our structural, biophysical and biochemical analyses demonstrate that the linker enhances the stability of GephE through specific interactions, particularly involving the $^{314}$RRHR$^{317}$ motif and suggest that the linker may play a regulatory role in gephyrin oligomerization by stabilizing the GephE domain by mediating intra- or intermolecular contacts.

### Cryo-EM structures of higher-order oligomeric states of GephFL

Having established the structural basis of 27F3 binding and the stabilizing role of the C-terminal linker in GephFL, we next investigated how gephyrin assembles into higher-order oligomers. Our focus was on the GephFL fraction 2 A′ which, based on SEC-MALS analyses (Fig. 1f), exhibits a molecular mass of ~350 kDa, consistent with a tetrameric assembly of GephFL.

To gain structural insights into the organization of these assemblies, we conducted a cryo-EM analysis. GephE could be clearly visualized, and the corresponding 2D classes revealed multiple distinct higher-order GephE oligomers, while representations of GephG were scarcely present (see below). The most prominent of the higher order gephyrin oligomers was a tetrameric assembly, in the form of a linear dimer of dimers, where subdomains II from different dimers mediate higher order interactions (Fig. 5a). In addition, we observed a hexameric population, organized as a linear trimer of GephE dimers (Fig. 5b), suggesting that gephyrin can form extended, linear oligomeric assemblies.

We also identified another distinct architecture, an oblique dimer of dimers, where the long axes of the GephE dimers intersect at an angle of approximately 145° (Fig. 5c). In this configuration, subdomain II of one dimer interacts with the core region of the second dimer. Among the three observed oligomeric species, the linear dimer of dimers was the most abundant, while the oblique arrangement was less frequently observed. Finally, we also observed a small population of only ~8500 particles, which displayed density for both GephG and GephE. Specifically, one GephG trimer was contacting one GephE monomer within a dimer via subdomain II (Fig. 5d). Upon examining

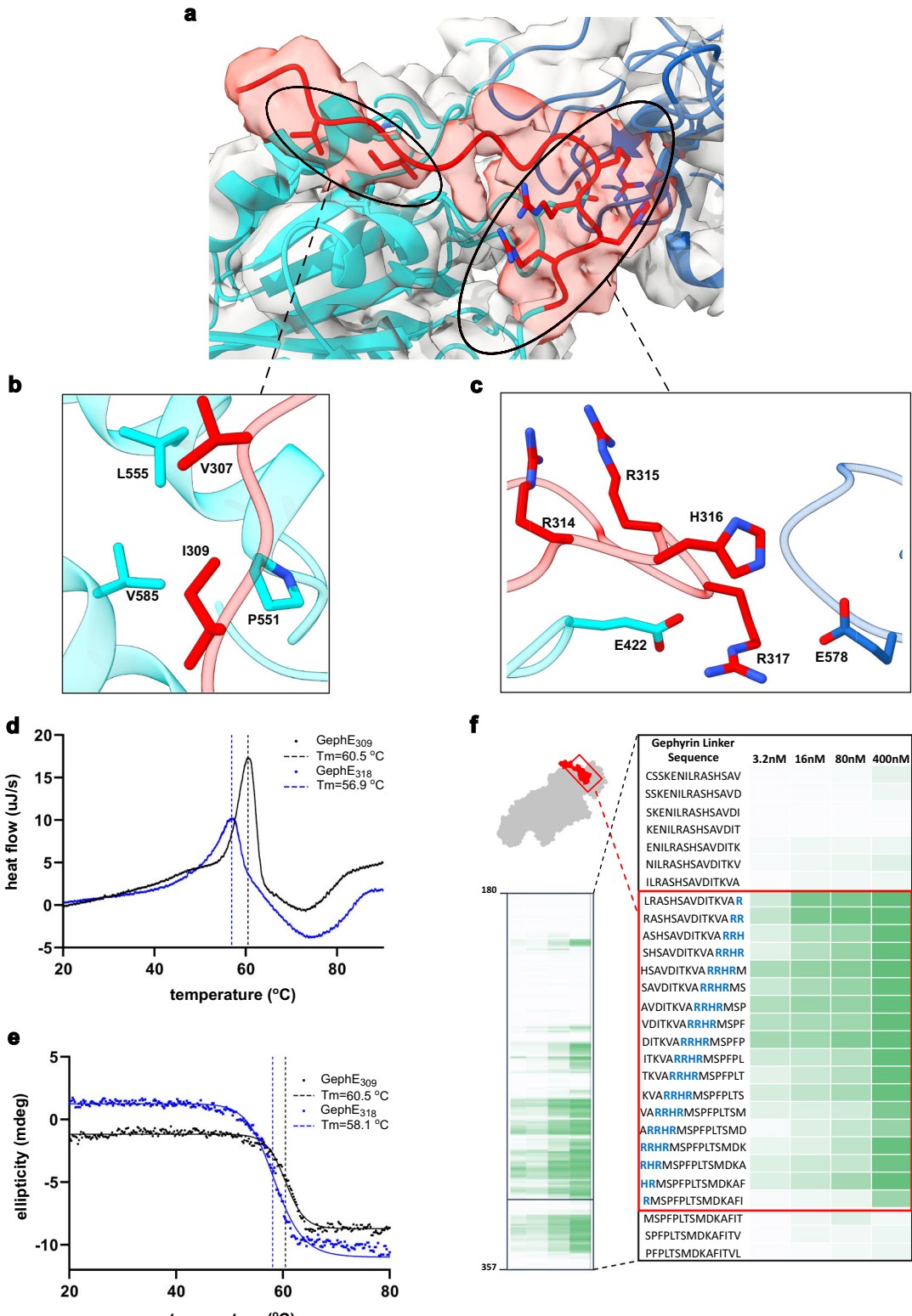

**Fig. 4 | Structure of residues 305–319 in the C-terminal part of the linker.**
**a** Density map showing residues 305-319 (red) and the adjacent residues in GephE.
**b**, **c** Close-up of the hydrophobic and electrostatic interactions between GephE and residues from the linker. **d** Unfolding of GephE318 and N-terminally elongated GephE309 as studied by DSC. **e** Unfolding of GephE318 and GephE309 as monitored by the change in the CD signal at 200 nm. DSC and CD unfolding experiments were performed in duplicates, results in d and e correspond to individual representative experiments. **f** Quantification of binding in peptide microarrays involving residues in the linker in the vicinity of the RRHR motif (highlighted in blue) observed to mediate electrostatic interactions. This region of the linker is one of several "hot-spots" identified to bind to GephE (left).

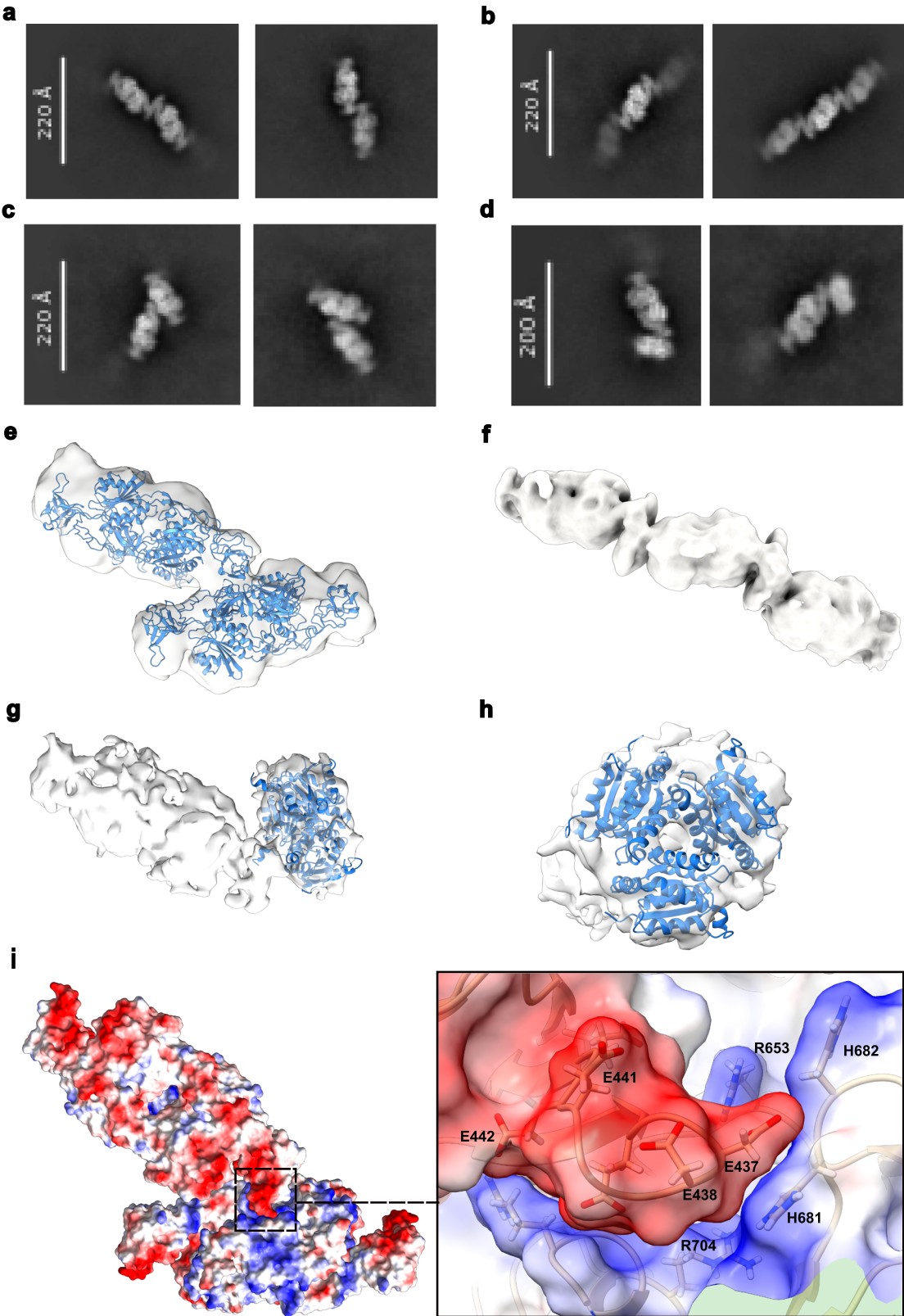

**Fig. 5 | Higher order oligomers of GephE and GephE-GephG interactions. a** 2D classes of the linear GephE dimer of dimers. **b** 2D classes of three GephE dimers arranged in a linear fashion. **c** 2D classes of the oblique GephE dimer of dimers. **d** 2D classes showing one GephG trimer contacting subdomain II of one monomer within a GephE dimer. **e** 3D volume of the oblique GephE dimer of dimers modeled with two dimers engaging as observed in the crystal structure of GephE in complex with a GlyR β-subunit receptor-derived peptide (PDB 4PD1). **f** Map of the linear arrangement of three GephE dimers. **g** Map of the GephG trimer-GephE dimer interaction with a model of the GephG trimer (blue, PDB entry 1JLJ) in the GephG density. **h** Density corresponding to the GephG trimer viewed along the 3-fold axis. The density for the GephE dimer has been omitted for clarity. **i** Electrostatic potential mapped onto a surface representation of the oblique dimer as observed in the crystal (PDB 4PD1) with electropositive regions and electronegative regions contoured at ± 10 kT/e⁻ in red and blue, respectively. Zoom into the interface between the negatively charged subdomain II of one dimer and the positively charged subdomain IV of the other dimer.

the oblique dimer arrangement, we noted that its interface resembled a crystal contact observed in the GephE-GlyR β-subunit peptide complex (PDB 4PD1)[35]. Hence, we positioned two GephE dimers, connected via this crystallographic contact, into the cryo-EM density, which yielded a good fit (Fig. 5e).

From the third population, we derived a low-resolution cryo-EM map of the hexamer, consisting of three GephE dimers arranged in a linear fashion, with the central dimer interacting with the flanking dimers via their subdomains II (Fig. 5f). Likewise, we could deduce a low-resolution map for the particles showing GephG and GephE in which GephG is bound asymmetrically to subdomain II of one GephE monomer with the other GephE subunit in the unbound state (Fig. 5g, h). The overall shape of the GephG trimer matches both the side view (Fig. 5g) and the view along the 3-fold axis of the GephG trimer (Fig. 5h).

Finally, additional information about the oblique dimer of dimers could be obtained from the GephE crystal structure (PDB entry 4PDI). The interaction between the two dimers is mediated by electrostatic complementarity (Fig. 5i) between subdomain II of one dimer and subdomain IV of the second dimer. Although the buried surface area is relatively small (925 Å²), and originally was not considered to be a functionally relevant interface, our cryo-EM data suggest it may in fact represent a physiologically meaningful oligomerization contact. The key residues involved in this interface include E386, E435, D437, D438, E441, E442 and E444 from subdomain II and K645, R653, H681, H682, R699 and R704 from subdomain IV. Despite the low resolution of this cryo-EM reconstruction, the consistency with known crystal contacts and the observed geometries imply that this interface may play a role in gephyrin assembly into higher-order oligomers via its E-domain.

## Structure of a linear dimer of GephE dimers

We proceeded with the structural analysis of the linear arrangement of GephE dimers, as a sufficient number of particles representing different orientations were available. Following image processing using CryoSPARC and RELION (Supplementary Methods and Supplementary Fig. 7), we obtained a cryo-EM map at a resolution of 3.1 Å (Fig. 6a). Model building and refinement were initiated by fitting two GephE dimers (PDB entry: 4PDO) into the density map. To distinguish the monomers within the linear tetramer, we designated them A through D, with monomers B and C mediating the interactions between adjacent GephE dimers.

During model building, it became immediately apparent that the interface between the two dimers is asymmetric (Fig. 6a, inset). Closer inspection also revealed significant conformational changes in subdomain IV, while subdomain II underwent rigid body movements. These motions of subdomain II have already been observed in the *E. coli* MoeA protein[36] and subsequently also in GephE itself. In the context of the linear dimer of dimers, these motions allow subdomains II from adjacent dimers to productively interact with each other. Subdomain II showed the lowest local resolution within the map, limiting our ability to confidently model side chains and certain loop regions. Nevertheless, we were able to identify the key interface regions mediating the dimer-dimer interaction. Two distinct contact sites appear to stabilize the assembly through electrostatic interactions. In the first region (Fig. 6b), R379 and D422 of monomer B engage in reciprocal interactions with R379, D422 and R452 of monomer C, forming the core of the interface between subdomains II regions. In the second region (Fig. 6c), R466 of monomer C appears positioned to form electrostatic interactions with D386 and E444 of monomer B. Notably, this interface is only observed on one side of the linear complex, as the

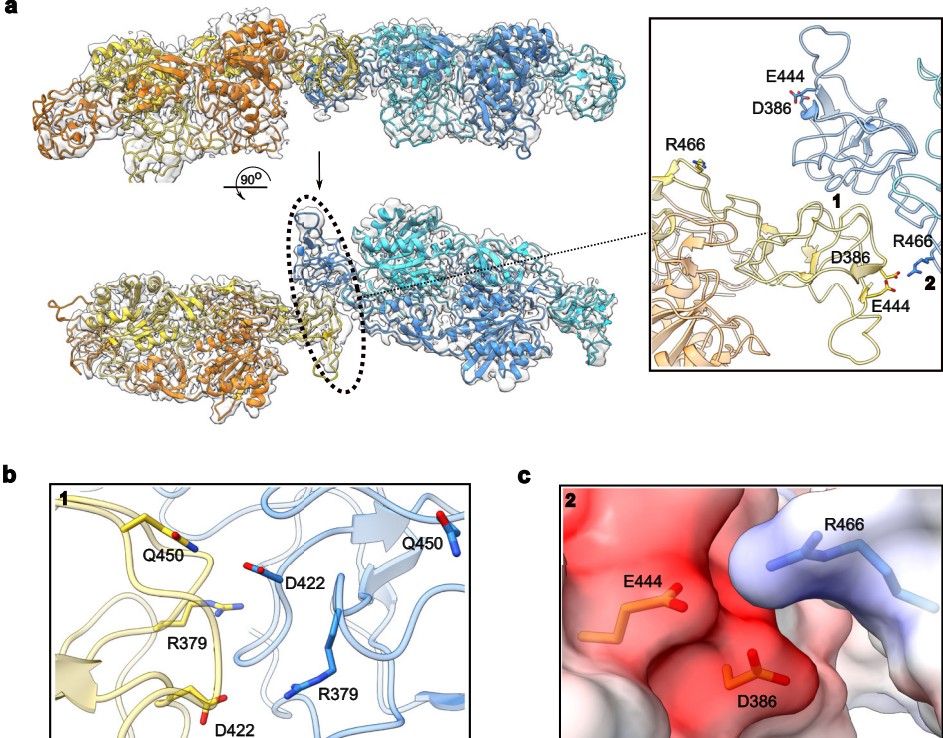

**Fig. 6 | Structure of the linear GephE dimer of dimers. a** Two orthogonal views of the dimer of dimers with the E-domains of two dimers in blue/cyan and in yellow/orange, respectively. Insert: Zoom into the interface between subdomains II (dark blue and yellow), depicting the asymmetry in the interaction. Arabic numerals 1 and 2 highlight regions where contacts stabilizing the tetramer are formed. **b, c** Residues contributing to the interface in regions 1 and 2, respectively. Region 2 is superimposed with an electrostatic potential map.

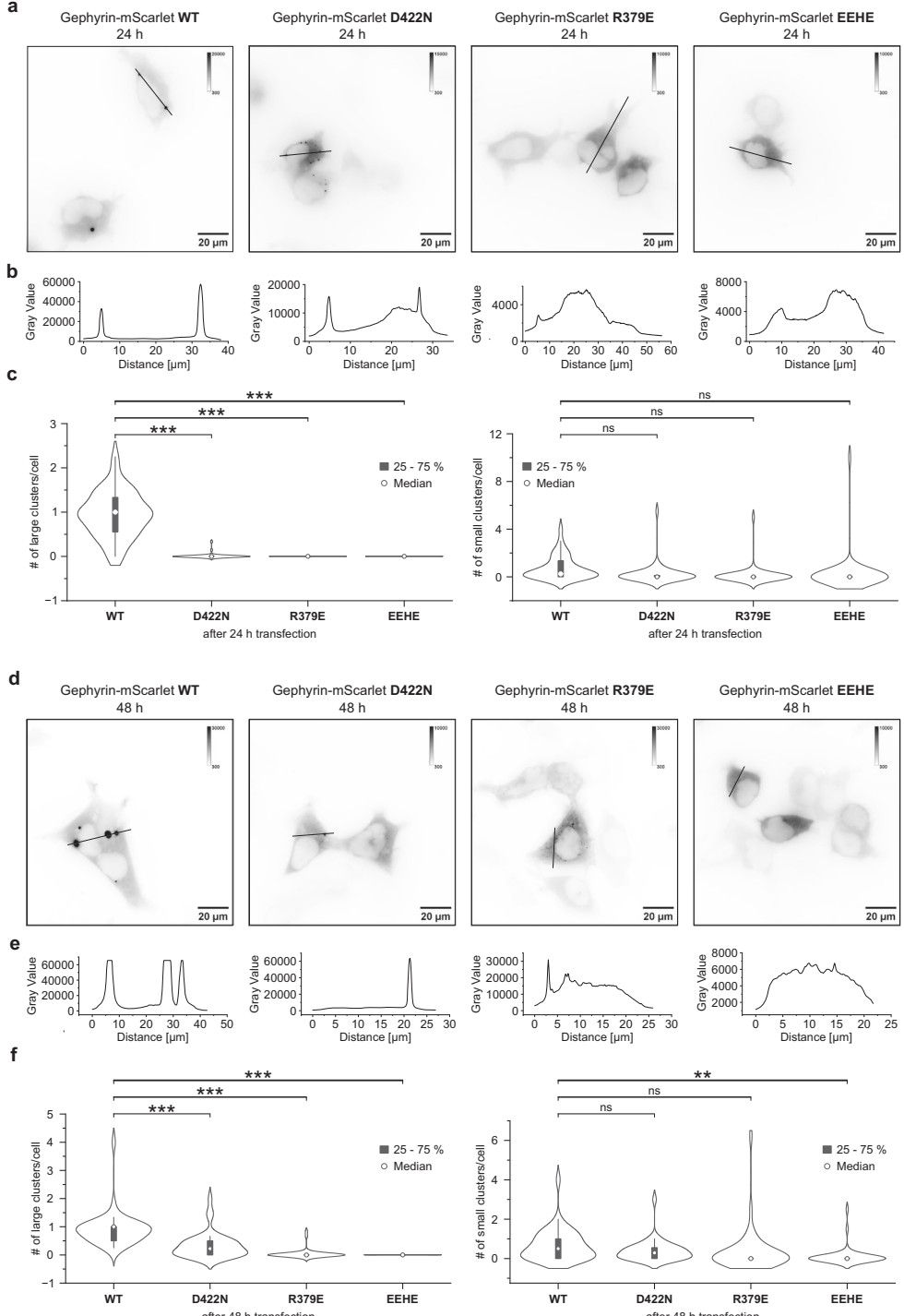

**Fig. 7 | Gephyrin condensate formation in HEK293T cells. a, d** Inverted representative wide-field fluorescence microscopy images of fixed HEK293T cells overexpressing mScarlet-labeled gephyrin WT, D422N, R379E and $^{314}$EEHE$^{317}$ variants. Cells were either transfected for 24 h (**a**) or 48 h (**d**) prior to fixation. **b, e** Line profiles along the black lines in the respective images in (**a**) and (**d**). **c** Cells were transfected for 24 h prior to fixation. Clusters with intensity > 25000 were identified as large clusters (left) and with intensity 5000–25000 as small clusters (right). Distribution of cluster counts per cell for the individual mutants is shown as violin plots. Each violin represents the kernel density estimate of the number of clusters per cell for the respective construct. The white dot indicates the median, the box denotes the interquartile range, and the thin line represents the rest of the distribution within 1.5 × the interquartile range. Individual data points (Supplementary Fig. S8) are overlaid to show cell-to-cell variability. One-way ANOVA for large

clusters: $F_{(3, 155)} = 133.8$, $p < 0.0001$. Tukey's HSD: all $p < 0.0001$. One-way ANOVA for small clusters: $F_{(3, 155)} = 2.3$, $p = 0.08$. Tukey's HSD: all $p > 0.05$. $n = 39$ (WT), $n = 40$ (D422N), $n = 40$ (R379E), $n = 40$ ($^{314}$EEHE$^{317}$) independent cells. **f** Cells were transfected for 48 h prior to fixation. Clusters with intensity > 50000 were identified as large clusters (left) and with intensity 20000–50000 as small clusters (right). Distribution of cluster counts per cell for the individual mutants is shown as violin plots (see (**c**)). Individual data points (Supplementary Fig. 8) are overlaid to show cell-to-cell variability. One-way ANOVA for large clusters: $F_{(3, 155)} = 48.4$, $p < 0.0001$. Tukey's HSD: all $p < 0.0001$. One-way ANOVA for small clusters: $F_{(3, 155)} = 3.5$, $p = 0.02$. Tukey's HSD: gephyrin WT vs $^{314}$EEHE$^{317}$ ≤ 0.01, gephyrin WT vs D422N and R379E $p > 0.05$. $n = 39$ (WT), $n = 39$ (D422N), $n = 40$ (R379E), $n = 40$ ($^{314}$EEHE$^{317}$) independent cells. Source data are provided as a Source Data file.

corresponding residues on the opposite side are too distant to engage in similar contacts (Fig. 6a).

## Electrostatic interactions are crucial for the formation of gephyrin condensates

Lee et al. demonstrated that deletion of gephyrin linker residues 286–326 disrupts gephyrin phase separation, observable as a lack of condensate formation in HEK293T cells[34]. The linker segment, encompassing residues 305–319, lies within this critical region and interacts with the oppositely charged subdomain II of GephE, which also contributes to the tetramerization interface identified in this study. To assess the role of residues identified in our structures, we tested how point mutations of key electrostatic interactions affect gephyrin condensation in cells (Fig. 7 and Supplementary Fig. 8). We introduced the R379E and D422N mutations and, in a separate construct, replaced the positively charged $^{314}$RRHR$^{317}$ motif (residues 314–317) with $^{314}$EEHE$^{317}$. D422N was chosen over D422R to avoid steric clashes and because it was identified in a patient with developmental and epileptic encephalopathy carrying a second truncating variant[37]. Wild-type (WT) gephyrin rapidly formed large intracellular puncta after 24 h (Fig. 7a, b), which further increased in size by 48 hours (Fig. 7d, e). R379E significantly delayed phase separation, with larger clusters only appearing after 48 h (Fig. 7d, e). D422N had a milder effect, with puncta forming already after

24 h (Fig. 7a, b) and some larger condensates appearing after 48 h (Fig. 7d, e). The $^{314}$RRHR$^{317}$/$^{314}$EEHE$^{317}$ substitution completely abolished clustering or slowed droplet fusion, similarly to R379E after 24 h (Fig. 7a, b) and 48 h (Fig. 7d, e), in line with prior evidence implicating the linker, especially the region of splice cassette C4c, as a hotspot for charge-mediated interaction essential to condensation. The formation of small and large clusters was quantified based on the intensity value and the cluster count per cell and was represented as violin plots for the respective variant after 24 h (Fig. 7c) and 48 h (Fig. 7f), further substantiating the effects of the mutants on condensate formation. Collectively, our data further refine the map of critical interactions within GephE and its adjacent linker, highlighting complementary contributions from both electrostatic interfaces and linker integrity to gephyrin condensate formation.

## Electrostatic interactions contribute to gephyrin clustering at inhibitory synapses

To assess the impact of the identified electrostatic interactions on gephyrin organization in neurons, we expressed mScarlet-tagged gephyrin variants (WT, D422N, $^{314}$EEHE$^{317}$ and R379E) in cultured hippocampal neurons by lentiviral infection at day in vitro (DIV) 12. The variants accumulated at inhibitory synapses to different degrees (Fig. 8a), as shown by their colocalization with the presynaptic marker

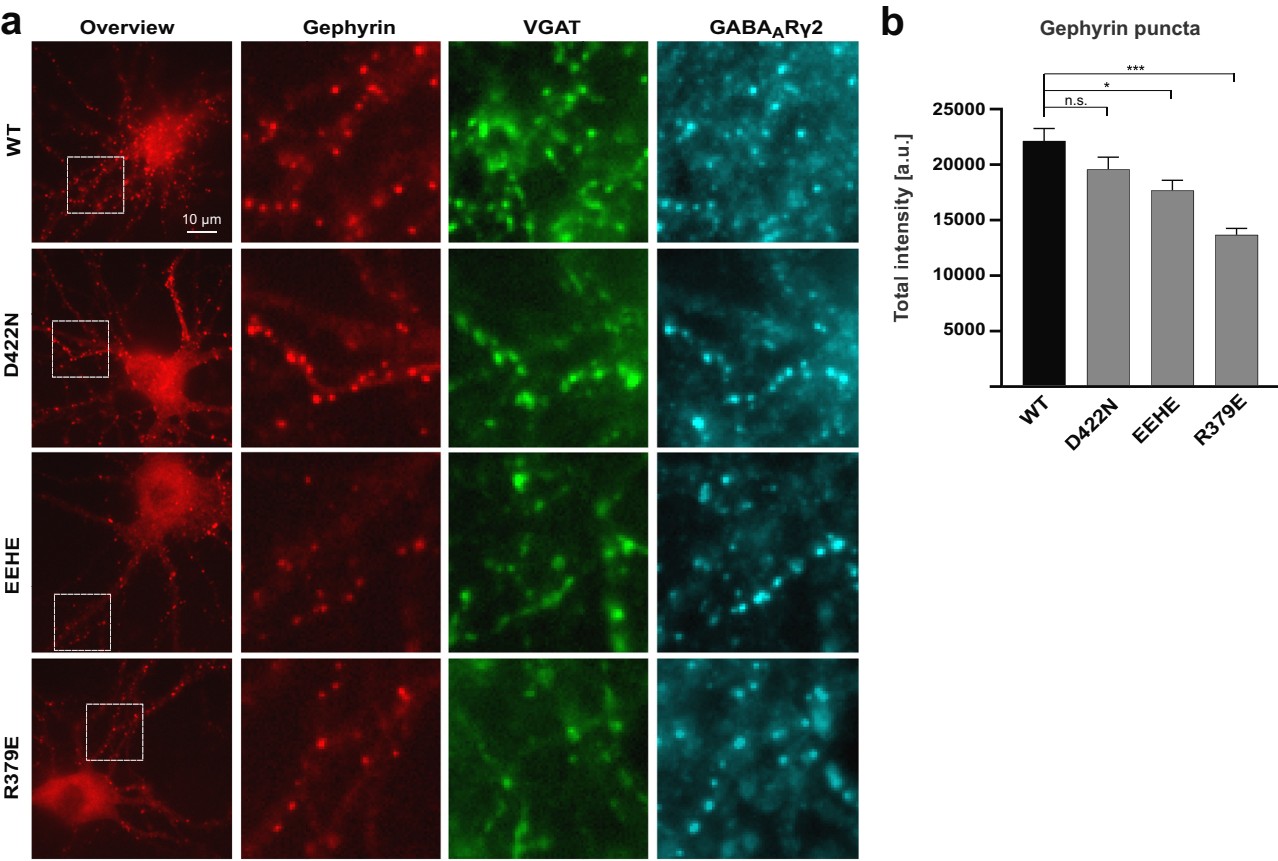

**Fig. 8 | Gephyrin clustering in cultured hippocampal neurons. a** Cultured hippocampal neurons infected at DIV12 with lentiviruses expressing gephyrin-mScarlet variants (red) were fixed at DIV20 and immunolabelled for the endogenous GABA$_A$R γ2 subunit (cyan) and the presynaptic marker VGAT (green). Each row shows representative images of hippocampal neurons expressing wild-type gephyrin (WT), as well as the variants D422N, $^{314}$EEHE$^{317}$ and R379E. Scale bar: 10 μm. Higher magnification of selected regions (white dashed squares, 20 μm side length) are shown on the right for all three channels. **b** Quantification of the total fluorescence intensity of synaptic gephyrin-mScarlet puncta in infected neurons. The integrated gephyrin-mScarlet fluorescence was measured at VGAT-positive

puncta, and the median value was calculated per cell ($n = 58$ cells for WT and $^{314}$EEHE$^{317}$, 57 cells for D422N, and 60 cells for R379E; mean ± SEM; KW test, *$p = 0.034$; ***$p < 0.0001$; not significant (n.s.) $p = 0.43$. The camera offset was corrected using the minimum pixel intensity in each channel. Data were pooled from four independent experiments ($N = 4$ biological replicates representing at least 8 coverslips per condition) to minimize variability arising from cell culture, labeling and imaging conditions, as well as the sampling of different neuron types. Similar results were obtained in all four experiments. Source data are provided as a Source Data file.

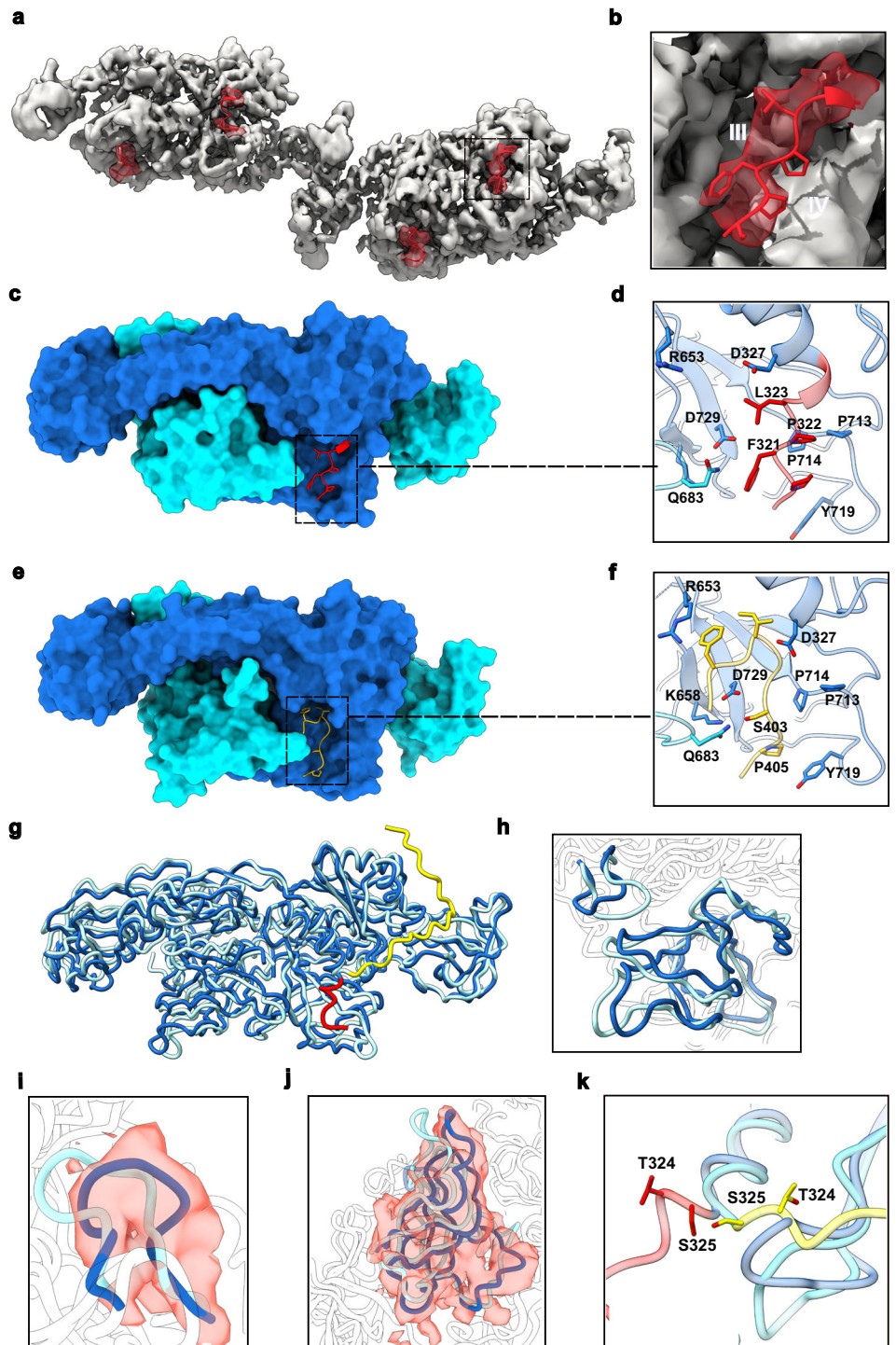

**Fig. 9 | Conformational changes observed in the linear GephE dimer of dimers.**
**a** Density map of the tetramer with residues 319–325 and the corresponding densities highlighted in red. **b** Zoom into residues 323–319 in the C monomer highlighting their proximity to subdomains III and IV. **c** Surface representation of the AB-dimer from the linear dimer of dimers with its subunits in blue and cyan, respectively. Residues 319–325 were excluded from the surface and are shown in stick representation in red. **d** Enlarged view highlighting the interactions between residues 319–325 (in red) and residues in the receptor-binding pocket (blue and cyan). **e** Surface representation of the GephE-GlyR β-loop structure (PDB entry 4PDO) with the GephE monomers in blue and cyan and the GlyR β-loop peptide in yellow. **f** Enlarged view highlighting the interactions between residues 397–411 from the β-loop peptide (yellow) and residues in the binding pocket (blue and

cyan). **g** Superimposition of the AB dimer (dark blue) from the linear GephE tetramer with the extended GephE-27F3 complex (light blue). For clarity, 27F3 is not displayed. Residues 305–325 from the linker in the GephE-27F3 complex are highlighted in yellow and residues 319–325 in the linear GephE dimer in red. **h** Enlarged view of the superimposition of subdomains III colored as in (**g**). **i** Conformational changes in residues 598–607 with the dimer of dimers map ("in conformation") in red, shown here for chain B. **j** Conformational changes in subdomain IV with the dimer of dimers map ("in" conformation) in red, shown again for chain B. **k** Close-up view highlighting the conformational changes in the vicinity of T324 and S325 (in stick representation). The "in" conformation is shown in red and the "out" conformation in yellow.

VGAT (green) and the GABA$_A$R γ2-subunit (cyan). This observation indicates that all variants can still be recruited to inhibitory synapses, and that disruptions of the critical interactions at the oligomeric interfaces do not abolish gephyrin's ability to associate with post-synaptic clusters. It should be noted that the neurons express endogenous gephyrin, meaning that the recombinant gephyrin variants are incorporated into existing synaptic gephyrin scaffolds.

To quantify the relative enrichment of the different variants at synaptic sites, we measured the total gephyrin-mScarlet intensity within VGAT-positive puncta (Fig. 8b). The D422N variant showed a small reduction in cluster intensity, which did not reach statistical significance. In contrast, the mean synaptic intensity of [314]EEHE[317] was modestly reduced compared to the wild-type protein ($p = 0.034$, Kruskal-Wallis (KW) test), whereas the GephR379E mutant showed a pronounced reduction relative to the wild-type ($p < 0.0001$, KW test) as well as to the other two variants ($p < 0.0001$ versus D422N and $p = 0.0043$ versus [314]EEHE[317]). These results match the trends observed in the HEK293T cell aggregation assays, in which [314]EEHE[317] and R379E also displayed the strongest impairment. Together, our data demonstrate that the identified interactions contribute to gephyrin clustering at synapses in a neuronal context.

### Conformational changes in the linker modulate receptor binding

Another striking feature of the linear GephE dimer of dimers structure is the orientation of residues 319–325, including F321 and L323, preceding the first α-helix. These residues form part of the N-terminal extension of GephE, which, in the GephFL-27F3 complex, wraps around subdomain III in a conformation we previously described as "out". In contrast, in the linear dimer of dimers structure, these residues adopt a distinct "in" conformation, consistently observed in all four monomers (Fig. 9a). A close-up view of this region shows F321 and L323 oriented toward subdomains III and IV (Fig. 9b).

Notably, in the "in" conformation residues 321–323 partially occlude the binding pocket for the GlyR and GABA$_A$ receptor-derived peptides, which is located at the interface of subdomains III and IV (Fig. 9c–f). In previous co-crystal structures, residues F398 and I400 from the GlyR β-subunit and F368 and I370 from the GABA$_A$R α3-subunit engage in hydrophobic interactions with gephyrin in this region. Figures 9d, f illustrate that although L323 does not precisely coincide with F398 or I400, it occupies the same hydrophobic patch. However, unlike the receptor-bound state, the intramolecular interaction between D327 and R653 remains intact. F321, which overlaps spatially with receptor residue S403, could engage in π–specific interactions rather than polar contacts. These interactions may include NH-π interactions with Q683, anion-π interactions with D729, and cation-π interactions with K658. Although these π-type interactions are generally weaker than polar contacts, their contribution to protein structural stability is well documented[35,36,38]. Similarly, L323 aligns with I400/I370 in the GlyR β/GABA$_A$R α3-derived peptides and engages in comparable hydrophobic interactions. In addition, P320 and P322 mimic contacts formed by I403 and P405 in GlyR β (or P368 and I370 in GABA$_A$R α3) by interacting with residues P713, P714 and Y719. These observations collectively suggest that conformational rearrangements are required to convert the gephyrin "in" conformation into the receptor-binding competent "out" state. As shown in Supplementary Fig. 9, the GephFL dimer of dimers is still able to bind the GlyR-β intracellular loop, showing that the "in" conformation does not prevent the interaction on its own. This switch would entail the replacement of relatively weak interactions with stronger contacts involving residues derived from the gephyrin-binding GlyR or GABA$_A$R subunits.

To further investigate this conformational plasticity, we superimposed the AB dimer from the linear tetramer onto the GephE-27F3 structure, where linker residues 305–325 are resolved. This comparison revealed conformational changes in both subdomains II and,

more prominently, in subdomain IV (Fig. 9g, h). These structural shifts were not restricted to a subset of monomers since all four GephE chains in the tetramer exhibited similar rearrangements, with pairwise RMS deviations ranging from 0.77–1.10 Å. The changes in subdomain IV can be described as a rotation of 6–9° away from subdomain III (Fig. 9h), depending on the monomer pair being compared. This rotation is coupled to a repositioning of the β-hairpin (residues 598–607) in subdomain III, leading to a displacement of P603 by more than 8 Å (Fig. 9i). These conformational changes are well supported by the cryo-EM density map (Fig. 9j). The divergence between the "in" and "out" linker conformations occurs at residues T324 and S325 (Fig. 9k), both of which are known phosphorylation sites[34,38]. This suggests a potential regulatory mechanism that modulates gephyrin's oligomeric state and receptor-binding competence via posttranslational modifications.

## Discussion

In this study, we provide a detailed structural framework for gephyrin assembly into higher-order oligomers. Using cryo-EM, we describe structural details of the gephyrin linker region, specifically its C-terminal segment, which was previously unresolved. This region engages in defined interactions with GephE, thus contributing to its structural stabilization. These interactions are driven in part by electrostatic complementarity and have been implicated in liquid-liquid phase separation (LLPS)[22,23]. Furthermore, their functional importance is underscored by our cellular aggregation assays, in which substitution of the [314]RRHR[317] motif completely abolishes gephyrin condensate formation in HEK293T cells. These results are further corroborated by analyzing the same variants in hippocampal neurons.

In parallel, we expand upon previous findings regarding the binding of the DARPin 27F3 to gephyrin by providing high-resolution cryo-EM structures of both GephE-27F3 and full-length gephyrin-27F3 complexes. These structures reveal that 27F3 binds exclusively to GephE, thereby engaging subdomains IV of both monomers through a combination of hydrophobic and polar interactions. Importantly, 27F3 binding is unaffected by the presence of GephG or the flexible linker, indicating that the entire epitope is entirely contained within GephE. These findings provide a molecular explanation for how 27F3 functions as a structural chaperone, enabling the stabilization and visualization of conformationally relevant states of gephyrin and its oligomeric assemblies.

A key discovery in this work is the intrinsic ability of GephE to form higher-order oligomers. Previous full-length structural models of gephyrin, specifically its P2 splice variant, were derived from small-angle X-ray scattering (SAXS) data generated by our lab[28]. In contrast to the P1 variant investigated in this study, the P2 variant contains 14 additional residues in the linker, which are inserted between residues 288 and 289. The resulting SAXS-based models featured a central core composed of a GephG trimer surrounded by flexible linker regions and GephE in a monomeric state in both compact and extended conformations. This mode of assembly clearly differs from the oligomerization behavior described here, where we find no evidence of monomeric gephyrin and instead observe defined higher-order assemblies of GephE dimers. In contrast, the SAXS-derived molecular masses supported a trimeric architecture for GephFL, which, in turn, was used as a $C_3$ symmetry constraint during model building. Our cryo-EM data challenge this assumption and instead point to a different organization of gephyrin in the P1 variant, emphasizing the importance of considering splice-specific effects on the supramolecular architecture of gephyrin.

Since SAXS based molecular mass determination is known to be sensitive to a high degree of flexibility and shape[37], as introduced by the flexible linker of gephyrin, masses for the gephyrin P2 variant were derived by several independent ways and agreed with GephFL being trimeric. Importantly, the ensemble optimization method employed to

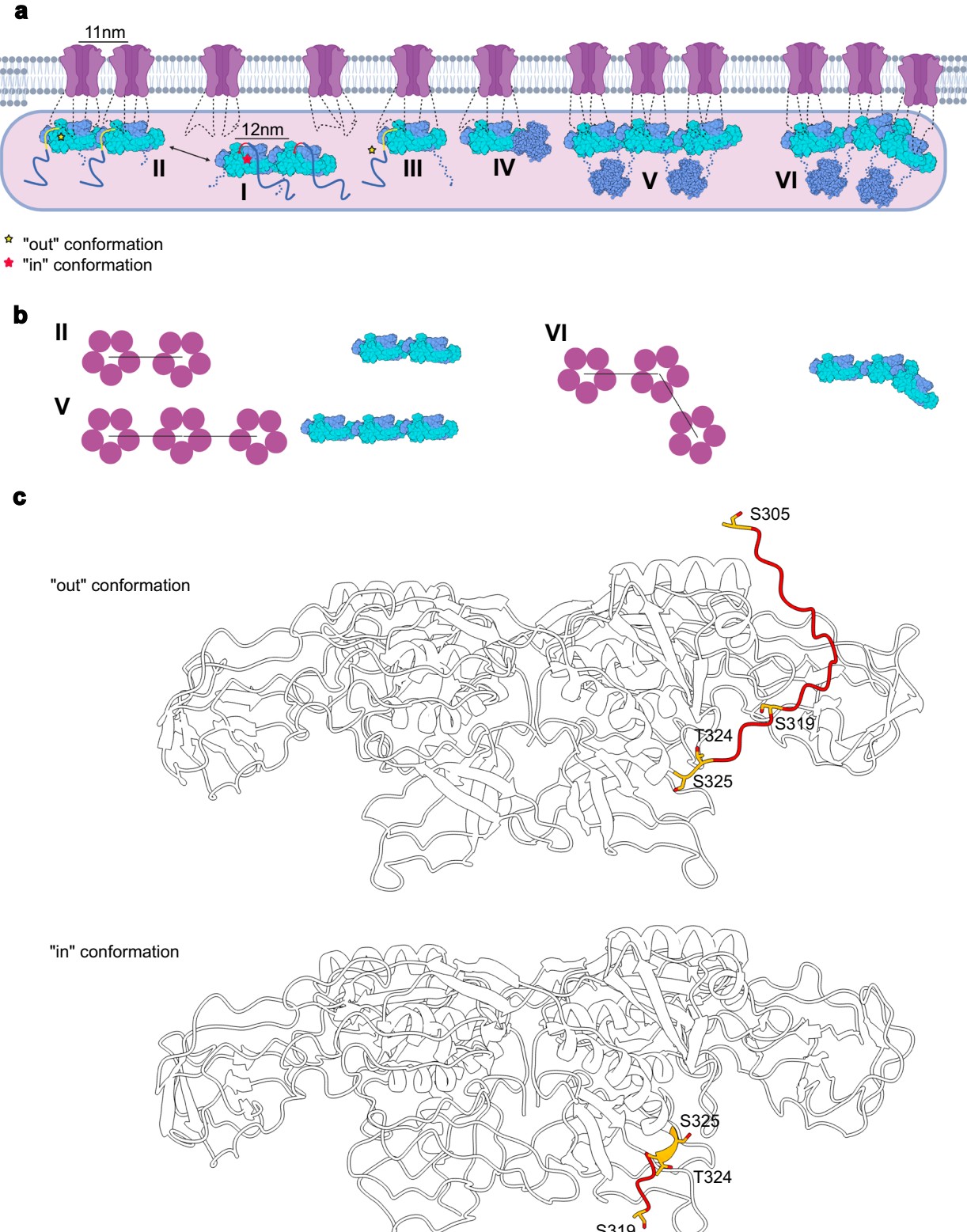

**Fig. 10 | Receptor clustering by different gephyrin oligomers and phosphorylation-dependent switch. a** A gephyrin dimer of dimers with the linker in the "in" conformation (I) is converted by binding to two receptors to a high-affinity binder by inducing the "out" conformation (II). The length of the dimer (vertical line above I) matches the receptor-receptor distance (vertical line above the receptors in II) as observed by cryo-ET[26]. The minimal receptor-gephyrin module (III) can be added to the dimer of dimers to either form a linear trimer of dimers (V) or by adding an oblique dimer to a dimer (VI). An additional oligomeric state (IV) corresponding to one GephE dimer contacting a GephG trimer via its subdomain II. With six gephyrin molecules, the formation of GephG trimers (3-fold symmetrical objects in V and VI) can be easily rationalized. This panel was partially created in Biorender; Ortiz-Lopez, D. https://BioRender.com/hh36phy **b** Models II, V and VI explain the receptor distribution observed by cryo-ET[26]. This panel was partially created in Biorender; Ortiz-Lopez, D. https://BioRender.com/9gr22jh. **c** Model of a gephyrin dimer with the linker in the "out" (top) and "in" conformation (bottom) with known Ser/Thr phosphorylation sites in stick representation. S305 could only be visualized in the "out" conformation.

analyze the SAXS data yielded a clearly superior fit when gephyrin was assumed to be trimeric compared to a dimer scenario. It should be noted that the gephyrin sample used for the SAXS studies was prepared following an established purification protocol and that this variant does not yield significant levels of higher-order oligomeric species (Supplementary Fig. 10). Thus, significant differences in the oligomerization behavior exist between the two isoforms. Nonetheless, we detected both dimeric and trimeric species by BN-PAGE for the P1 variant, supporting a dynamic equilibrium and rapid transitions between oligomeric states, consistent with predictions from Monte Carlo simulations[22].

Lee et al. [22] previously demonstrated a strong predictive relationship between the ability of gephyrin to undergo phase separation in heterologous cell systems and its synaptic clustering, synaptogenesis, and functional integration into inhibitory neuronal networks. Our cryo-EM structure of the linear GephFL dimer of dimers identifies charged residues in subdomain II, specifically R379 and D422, as critical mediators of electrostatic interactions between adjacent dimers (Fig. 7). Mutation of these residues disrupted condensate formation in HEK293T cells, with the R379E charge-reversal mutation exhibiting a more severe phenotype than the conservative D422N substitution. The D422N variant, originally identified in a patient with developmental and epileptic encephalopathy[39], had previously been reported to impair synaptic clustering. Interestingly, in vitro studies suggested that D422N induces an altered 3D structure, which is somewhat unexpected given the conservative nature of the substitution. Our findings suggest a more plausible explanation: the D422N mutation does not grossly alter tertiary structure and instead weakens critical quaternary interactions between GephE dimers within the linear assembly. Thus, the interfaces resolved in our study are likely essential not only for phase separation but also for gephyrin-mediated receptor clustering, synapse formation, and overall inhibitory synapse function. The phenotype of the R379E variant was further confirmed by studies in hippocampal neurons, in which integration of the variant in existing synaptic scaffolds was strongly disrupted.

In addition, the structure of the linear GephE dimer of dimers revealed substantial conformational rearrangements in subdomain IV (residues 656–732), which influence the access to the receptor-binding pocket. Residues within the linker adopt the "in" conformation, thus obstructing the pocket. Prior crystal structures lacked these linker residues entirely, leaving the binding site freely accessible. In the GephFL-27F3 complex, where portions of the linker are resolved, these residues adopt the "out" conformation, rendering the site accessible for receptor engagement. The divergence points between the "in" and "out" conformations lie near T324 and S325 residues that are subject to phosphorylation.

Several phosphorylation sites have been identified within this linker region, including S305[40] and S325[38], which are substrates for Ca$^{2+}$/calmodulin-dependent protein kinase II (CaMKII) as well as S319[41] and T324[34]. This raises the possibility that the "in" versus "out" conformational states are dynamically regulated by kinase and phosphatase activity. Support for this model comes from studies showing that phosphorylation of S325 promotes synaptic GlyR accumulation in Drosophila Mauthner cells[38]. Thus, post-translational modifications in the linker may serve as a molecular switch that modulates gephyrin's receptor-binding competence, its synaptic targeting, and the functional assembly of inhibitory postsynaptic domains.

The in situ organization of GABA$_A$ receptors at inhibitory synapses was reconstructed using cryo-ET[26]. Despite notable heterogeneity in receptor-gephyrin assemblies, a characteristic inter-receptor distance of approximately 110 Å was consistently observed (Fig. 10, vertical line above model II). This value closely matches the ~123 Å separation between corresponding atoms of adjacent GephE dimers in our linear GephE dimer of dimers structure (Fig. 10, vertical line above model I).

Moreover, among receptor pairs separated by this characteristic distance, the most common arrangement was that of a single neighbor, consistent with a linear organization of receptor dimers, each tethered by a single GephE dimer (Fig. 10, model II). Notably, formation of such an assembly requires displacement of the linker region from the "in" conformation, allowing receptor loops to occupy the gephyrin-binding pocket (Fig. 10, model II). While the resolution of the cryo-ET reconstructions was limited, two out of four receptor-scaffold assemblies displayed density features attributable to the scaffold, which closely resemble our linear GephE dimer of dimers.

The second most prevalent receptor organization identified in the cryo-ET study was a non-linear trimeric configuration[26]. This arrangement can be explained by combining the linear GephE dimer of dimers with the oblique dimer of dimers also observed in our cryo-EM analysis (Fig. 10, model VI). Additional receptor-gephyrin configurations are also plausible, including isolated receptors in complex with gephyrin dimers (Fig. 10, model III), linear chains of three receptors each bound to distinct GephE dimers (Fig. 10, model V) and a GephE-GephG direct interaction via GephE subdomain II (Fig. 10, model IV).

The regulation of GephFL oligomerization represents an important aspect for further discussion. As shown in Fig. 1a, our size-exclusion chromatography experiments reveal a dimer of dimers peak even at low protein concentrations, indicating that this oligomeric state is stable and not a consequence of high protein concentration. In contrast, a recent study by Macha et al. [42] reported oligomerization of the isolated GephE domain only at high concentrations, suggesting that additional regions of gephyrin may contribute to stabilizing higher-order assemblies. In this context, our data suggest that GephG and/or the linker play a role in stabilizing interactions between GephE dimers. Importantly, we further report a low-resolution cryo-EM map that provides direct evidence of an interaction between GephE subdomain II and the GephG trimer. This map lacks the dimer of dimers arrangement, suggesting that binding of the G domain may modulate the oligomeric state of gephyrin and potentially regulate higher-order assembly.

In summary, our cryo-EM structures of gephyrin oligomers provide a direct molecular basis for the higher-order architectures observed at inhibitory postsynaptic densities. These oligomeric arrangements appear to underlie the spatial organization of GABA$_A$Rs in vivo, supporting a model in which gephyrin scaffolds dynamically assemble into structurally defined, functionally relevant synaptic platforms.

## Methods

### Plasmid constructs, protein expression and purification

A full list containing all plasmids used in this work is provided in Supplementary Table 1. Proteins were expressed in *Escherichia coli* BL21 cells. Expressions were carried out overnight at 15 °C following induction with 0.5 mM isopropyl-β-D-1-thiogalactopyranoside. GephE$_{318}$ (residues 318–736) was purified as described[43]. GephE$_{309}$ (residues 309–736) was purified using Ni$^{2+}$-NTA affinity chromatography (50 mM Tris-HCl pH 8.0, 500 mM NaCl, 10 mM imidazole, 10% glycerol, 5 mM β-mercaptoethanol), followed by ion exchange chromatography (20 mM Tris-HCl pH 8.0, 250–500 mM NaCl, 1 mM EDTA, 5 mM β-mercaptoethanol) using a Mono Q 10/100 GL and size exclusion chromatography with a Superdex 200 increase 10/300 GL column (Cytiva) in 20 mM Tris-HCl pH 8.0, 250 mM NaCl, 1 mM EDTA and 5 mM β-mercaptoethanol). To study the interaction between higher-order oligomers of gephyrin and GlyRs, we generated a construct containing the GlyR β-subunit intracellular loop (residues 208–341) fused to coiled-coil repeats of the GCN4 transcription factor (referred to as GlyR-βGCN4), a similar construct as described[22] and cloned it into the pETM11 vector.

His-tagged DARPin 27F3 was purified by Ni-affinity chromatography using an equilibration buffer containing 50 mM Tris, pH 8.0, 300 mM NaCl, and 10 mM imidazole. Two wash steps at low (100 mM NaCl) and high (500 mM NaCl) salt concentrations were performed with the same buffer. Proteins were eluted with 50 mM Tris, pH 8.0,

300 mM NaCl and 250 mM imidazole. Eluted fractions were pooled and further purified by size exclusion chromatography on a Superdex 75 Increase 10/300 GL column (Cytiva) equilibrated with 50 mM Tris, pH 8.0, and 150 mM NaCl.

GephFL splice variant P1 was purified by Ni-affinity chromatography in 50 mM HEPES, pH 8.0, 300 mM NaCl, 10% glycerol, and 5 mM β-mercaptoethanol. Washes were performed with the same buffer supplemented with 10 mM and 30 mM imidazole. Proteins were eluted with 50 mM HEPES, pH 8.0, 50 mM NaCl, 5 mM β-mercaptoethanol, and 250 mM imidazole. The eluate was applied to a Mono Q 10/100 GL column and eluted over a 30-column volume gradient from 0 to 1 M NaCl in 20 mM HEPES, pH 8.0, 1 mM EDTA, and 5 mM β-mercaptoethanol. To maximize oligomeric purity, the protein was further purified by two rounds of size-exclusion chromatography on a Superose 6 Increase 10/300 GL column (Cytiva) equilibrated with 20 mM HEPES, pH 8.0, 150 mM NaCl, 1 mM EDTA, and 5 mM β-mercaptoethanol.

## SDS-PAGE and blue native PAGE
Proteins were analyzed by SDS-PAGE using 15% polyacrylamide gels. Samples were reduced with sample buffer containing 5 mM β-mercaptoethanol and were heated at 95 °C for 5 min before loading. Electrophoresis was run at 120 V, and bands were visualized with Coomassie staining. 2 µg of protein were loaded per well.

Blue native PAGE analysis followed the protocol developed by Schaegger et al. [44]. Samples were diluted in sample buffer (75 mM Bistris, pH 7.4, 10% glycerol) and loaded in a precast gel NativePAGE™ Novex® 3–12% Bis-Tris gel (Invitrogen). Electrophoresis was performed on an XCell™ SureLock™ Mini-Cell at a voltage of 150 V.

## SEC-MALS measurements
For molecular weight determination, gephyrin samples were analyzed by SEC-MALS. For each sample, 100 µL were injected at concentrations of 1.3 mg/mL for A1, 1.3 mg/mL for A2′, and 1.2 mg/mL for B1′, using the same buffer conditions as in the preparative size-exclusion chromatography step. Separations were performed on a Superose 6 Increase 10/300 GL column (Cytiva) connected to an ÄKTA pure system (Cytiva) at a flow rate of 0.5 mL/min.

The chromatography system was coupled to a multi-angle light scattering (MALS) setup (Wyatt Technology) consisting of a DAWN HELEOS 8 + light scattering detector and an Optilab T-rEX refractive index detector. Light scattering was recorded at multiple angles using a laser wavelength of 663.3 nm, and the protein concentration was determined from refractive index measurements. Molecular weights were calculated using the Zimm model with a dn/dc value of 0.185 mL/g. Detector normalization and band-broadening corrections were applied following the manufacturer's recommendations. Light scattering and concentration data were integrated to calculate molecular masses at one-second increments using the Astra 6.1.5 software (Wyatt Technology).

## CryoEM sample preparation
The following samples were prepared: GephFL(dimer)-27F3, GephE_{309}(dimer)-27F3, GephFL(dimer of dimers). As a first step prior to vitrification, 27F3 in 20 mM Hepes, pH 8.0, 150 mM NaCl, 5 mM β-mercaptoethanol and 5 mM EDTA, was incubated in a 1:1 molar ratio with GephFL or GephE_{309}, respectively, at 4 °C. The same conditions were also applied for the oligomeric sample of GephFL in the absence of 27F3. All proteins were subsequently subjected to size exclusion chromatography on a Superose 6 10/100 column equilibrated with the incubation buffer to eliminate aggregates and to obtain monodisperse samples.

Vitrification was carried out in a Vitrobot Mark IV (Thermo Fisher) at 4 °C and 100% relative humidity. Each blotting was performed with 3.0 µL of sample at a concentration of 0.1 mg/ml applied on glow-discharged R1.2/1.3 carbon grids (Quantifoil) using a blotting force of 20 for 5 s. The grids were subsequently plunge-frozen in liquid ethane. Data collections were performed at the CM01[45] and CM02 beamlines of the European Synchrotron Radiation Facility (ESRF) and the cryo EM-facility of the Julius Maximilians University Würzburg. Details regarding each data collection are available in Supplementary Table 2.

## CryoEM processing and model building
Data processing was carried out using CryoSPARC 4[46], and for the specific case of the GephFL(dimer of dimers) map, data were transferred to Relion 5.0[47] using PyEM 0.5[48] to perform 3D autorefine and multibody refinement steps. Further details regarding the processing of each dataset are described in Supplementary Figs. 3, 4 and 7.

For each map, the PDB coordinates were manually docked into the maps using the PDB entry 4PD0 for GephE and, where applicable, a model generated with AlphaFold3[43] for 27F3. The initial models were improved by iterative rounds of manual model building using Coot[49] and real space refinement with Phenix[50] against maps sharpened with Phenix (GephFL structures) or the unsharpened map (GephE_{309}-27F3 complex). Final model statistics are available in Supplementary Table 2.

## Differential scanning calorimetry (DSC)
Differential scanning calorimetry (DSC) experiments were performed using a *Nano DSC* instrument (TA Instruments). GephE_{309} and GephE_{318} were analyzed at concentrations of 0.8 and 0.6 mg/ml, respectively, along with a buffer-only baseline control containing 20 mM HEPES, pH 8.0, 150 mM NaCl and 5 mM β-mercaptoethanol. Each thermogram was corrected by subtracting the blank signal. The melting temperature ($T_m$) was defined as the temperature corresponding to the maximum of the corrected heat flow curve with GraphPad Prism (Dotmatics, Boston).

## Circular dichroism spectroscopy
Circular dichroism (CD) thermal denaturation experiments were performed using a JASCO J-810 spectropolarimeter equipped with a Peltier temperature control element. Samples included the GephE_{309} and GephE_{318} variants at a concentration of 0.2 mg/ml in 50 mM potassium phosphate buffer. Thermal unfolding was monitored at 200 nm, a wavelength sensitive to overall secondary structure, including disordered regions, from 20 °C to 90 °C in 0.2 °C increments. Ellipticity values were fitted to a two-state Boltzmann sigmoidal model using non-linear regression in GraphPad Prism. The melting temperature ($T_m$) was defined as the temperature corresponding to the maximum of the first derivative of the fitted curve. To compare the secondary structures of GephE_{309} and GephE_{318}, far UV CD spectra were recorded between 195–260 nm.

## High density peptide microarray analysis
Peptide microarray analysis was performed using high-density peptide microarrays synthesized by Schafer-N ApS (Copenhagen, Denmark) as previously described[51]. The entire linker region of gephyrin (P1 variant) was displayed as 12 duplicates over 12 sectors, each consisting of 5187 individual peptide fields. Gephyrin residues 180–357 were synthesized as 177 overlapping peptides (15 amino acids in length), with each peptide overlapping by 14 amino acids (offset by one amino acid) across all 12 sectors. As control for synthesis quality and as reference markers for sector alignment, 20 to 100 FLAG peptides (DYKDDDDK) were included in the four corners of each sector.

For binding studies, Alexa-647 labeled GephE was used in assay buffer consisting of 130 mM NaCl, 50 mM Tris-acetate (pH 8.0) and 0.1% Tween-20. The different sectors of the microarrays were incubated with fluorescent GephE at concentrations of 3.2 nM, 16 nM, 80 nM, and 400 nM for 1 h at room temperature. Following incubation, the microarrays were washed for 1 h with assay buffer. After air-drying of the slides in a nitrogen jet, the microarrays were scanned using an

Innoscan 900 laser scanner (Innopsys, Carbonne, France) at a resolution of 1 μm. The resulting images were analyzed using the PepArray analysis program (Schafer-N, Copenhagen Denmark). Auxiliary marker peptides with sequences containing the FLAG epitope tag were used for positioning the grid and to quantify field intensities. Binding intensities were measured for each peptide field, and these intensities were normalized to the highest intensity of binder within each sector for consistent comparison across microarray sectors.

## Gephyrin clustering in HEK293T cells

HEK293T cells were obtained from the German Collection of Microorganisms and Cell Cultures (DSMZ). The cells were cultured in DMEM (GIBCO), supplemented with GlutaMax and pyruvate (GIBCO), 10% fetal bovine serum (GIBCO) and 1% penicillin/streptomycin (Sigma) at 37 °C under 95% $O_2$/5% $CO_2$. The cells were plated on 4-chambered #1.5 high-performance cover glass slides coated with 35 μg/ml poly-D-lysine and were transfected with either WT Geph-mScarlet or a mutant. The transfection was performed at 60–80% confluency. Shortly before transfection, the medium was changed to fresh DMEM. 0.7 μg plasmid DNA was added to 70 μl JetPrime buffer, vortexed for 10 s, and spun down. 1.4 μL JetPrime reagent were added, mixed immediately and incubated for 10 min at RT. The transfection mix was pipetted dropwise on cells while swirling for 24 h. The following day, the cells were used for cell fixation and widefield fluorescence microscopy.

Coverslips with HEK293T cells were fixed in 0.1 M sodium phosphate buffer, pH 7.4 containing 4% paraformaldehyde (EM grade, Polysciences) and 1% sucrose for 20 min at 37 °C, rinsed three times with PBS and imaged in PBS. The measurements were taken from distinct samples with a sample size ≥2 for each group. A series of images, used to generate the data points, was acquired from different regions of the sample, each region having a distinct group of cells. The samples were imaged on an inverted Leica DMI6000B microscope with a 100 x oil-immersion objective (NA 1.49) using a Leica DFC9000 GTC VSC-05760 sCMOS camera (16-bit, 2 × 2 binning, image pixel size: 130 nm). A minimum of 40 images were acquired at a frame rate (exposure time) of 100 ms and constant illumination intensity to ensure comparability. Image processing and analysis were carried out using Fiji[52]. For cluster quantification, a contrast of either 300-26000 (24 h transfection) or 300–50000 (48 h transfection) was applied. After 24 h of transfection, clusters with intensity values 5001–26000 were identified as large clusters, ≤ 5000 as small clusters. After 48 h of transfection, clusters with intensity values 20001–50000 were identified as large clusters, ≤ 20000 as small clusters, and the cluster count per cell was determined for the individual images.

## Gephyrin clustering in hippocampal neurons

Experiments were conducted in accordance with the European directive on the protection of animals used for scientific purposes (2010/63/EU) and local veterinary regulations (Inserm UMS44-Bicêtre, license G94043013). Lentiviruses were produced as described[53] using the mScarlet-tagged gephyrin WT and the [314]EEHE[317], R379E and D422N variants (pLVX-mScar-gephyrin-IRES-ZsGreen1 constructs). Hippocampal neurons were isolated from E17.5 Swiss mouse embryos. Hippocampal tissue was dissected in cold HBSS with 20 mM HEPES, trypsinized (37 °C, 15 min), and triturated in MEM containing 0.3 mg/mL DNase I and 10% heat-inactivated horse serum. Neurons were plated on poly-D,L-ornithine-coated coverslips ($3.2 \times 10^4$/cm$^2$) and maintained in neurobasal medium supplemented with B-27 and 2 mM GlutaMAX. The medium was partially replaced once a week. Neurons were infected with lentivirus at DIV12 and fixed at DIV20 in 100 mM phosphate buffer pH 7.4 containing 4% PFA and 1% sucrose (10 min).

For immunolabelling, neurons were permeabilized and blocked in PBS with 0.25% Triton X-100 and 4% BSA, and incubated with primary antibodies against the $GABA_AR$ γ2 subunit (rabbit polyclonal, Synaptic Systems #224003, 1:500) and VGAT (guinea pig polyclonal, Synaptic

Systems #131004, 1:500) for 2 h, followed by donkey anti-guinea pig AF488 (Jackson ImmunoResearch #706-545-148, 1:1000) and donkey anti-rabbit AF647 secondary antibodies (Invitrogen #A-21246, 1:1000). Images were acquired on an ELYRA PS.1 microscope (63x/NA 1.4 oil, EMCCD camera, 250 nm pixel size) in the green, red, and far-red channels. Synaptic puncta were detected in the VGAT channel to create a mask[53] using the Icy spot detector (scale 2, sensitivity 80, 4–100 px size). The integrated intensity of the puncta was then measured in the red mScarlet channel and was thresholded using the mean intensity + 2x SD (standard deviation) of non-infected control neurons. Intensity values were also corrected for camera offset (1488 a.u.). Data are presented as mean ± SEM; statistical significance was assessed using a Kruskal-Wallis (KW) test with a Dunn's post-hoc test.

## Reporting summary

Further information on research design is available in the Nature Portfolio Reporting Summary linked to this article.

## Data availability

Unless otherwise stated, all data supporting the results of this study can be found in the article, supplementary, and source data files. The structural data generated in this study were deposited in the Electron Microscopy Database (EMDB) and the Protein Data Bank (PDB) with the following accession codes: 9S3M, EMD-54540 for the $GephE_{309}$-27F3 complex, 9S3F, EMD-54532 for the GephFL-27F3 complex and 9S3T, EMD-54551 for the GephFL dimer of dimers. The microscopy data generated in this study for clustering experiments in both HEK293T cells and in hippocampal neurons were deposited in figshare [https://doi.org/10.6084/m9.figshare.31676059]. Source data are provided in this paper.

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

## Acknowledgements

We kindly acknowledge the support of the personnel from the CM01 beamline at the ESRF, including Dr. Eaazhisai Kandiah and Dr. Alessandro Grinzato, for giving access, support and perform experiments at their facility. In addition, we also acknowledge the ESRF for the provision of beam time on the CM02 CRG beamline. The purchase of this microscope was funded by the EquipEx+ France Cryo-EM project (ANR-21-ESRE-0046). We thank Dr. Gregory Effantin and Dr. Pauline Juyoux for performing the experiments and providing local support, and Dr. Guy Schoehn for the establishment of this cryoEM facility. We kindly acknowledge the cryoEM-facility of the Julius-Maximilians University Würzburg (supported by DFG grants 359471283, 456578072 and 525040890) and Christian Kraft for data collection. We thank Dr. Jochen Kuper for the critical reading of the manuscript and insightful discussion.

 

This study was supported by DFG grant 232550447 to Hermann Schindelin, DFG grant MA6957/2-1 to Hans Maric, the University of Zürich Forschungskredit Can-doc grant to Benjamin Campbell, and the SNSF grant 310030_192522 to Shiva Tyagarajan.

## Author contributions

D.O.L, H.M.M., and H.S. designed the study. D.O.L, T.T.H., and H.S. performed the CryoEM data processing and model building. Biochemistry experiments (protein purifications, DSC, CD, SEC-MALS) were performed by D.O.L. with relevant contributions of P.M.v.g.H, B.F.N.C., and B.S. Peptide array data were generated by H.M.M. Experiments in HEK293T cells were designed and performed by C.H. and H.M.M. Experiments in hippocampal neurons were designed and performed by S.C. and C.S. The manuscript was written by D.O. and H.S. with relevant feedback and contributions from A.P., S.T., H.M.M., B.F.N.C., and B.B.

## Funding

## Competing interests

The authors declare no competing interests.
