## [Transparent Peer Review file · Nature Communications]

Cryo-EM Structures of Higher Order Gephyrin Oligomers Reveal Principles of Inhibitory Postsynaptic Scaffold Organization

Corresponding Author: Professor Hermann Schindelin

Version 0:

Reviewer comments:

Reviewer #1

(Remarks to the Author)

In this study, the authors investigated the structure and assembly mode of the full-length Gephyrin (Geph) by combining biochemical and cryo-EM studies, and elucidated the dimerized form of the full-length GPHN, which differs from previous studies suggesting the hexagonal assembly of Geph. Notably, the biochemical purification of recombinant full-length Geph from *E. coli* demonstrated the heterogeneous assembly modes of Geph, and the Cryo-EM analysis of each fraction of Geph provided a deep mechanistic understanding of dimerized and higher-order oligomerized Geph. The new model of Geph assembly derived from this study is well-aligned with the Cryo-ET observation of iPSD in situ and suggests potential regulatory mechanisms of Geph assembly in neurons. The overall structural information is valuable for understanding iPSD assembly formation and regulation. This is a strong candidate for NC pending some relatively minor revisions.

Major points:

- (1) The use of DARPin-27F3 for visualizing GephE and the dimerized GephFL structure is interesting. Based on the structure, the authors discussed the influence of DARPin-27F3 binding on the GlyR β -GephE interaction. However, since the interaction between GlyR β -loop and GephE is quite strong, the potential competition between DARPin-27F3 and GlyR for Geph binding needs more detailed biochemical analysis. This is a good opportunity to do this, as such characterization will also add value for using DARPin-27F3 in iPSD visualization in neurons as well as in other applications.
- (2) According to the fractionation and structural analysis of the full-length Geph demonstrated in this paper, there exist multiple higher-order assemblies of GephFL mediated by the interaction between subdomain II from two GephFL dimers. But the trimerized G-domain could not be observed by Cryo-EM. Is the G-domain trimerization needed for the higher-order assembly of GephFL? Would GephE alone form larger oligomers at high concentrations?
- (3) The Cryo-EM analysis of GephFL dimer and tetramer showed two distinct conformations of the C-terminal linker regions with opposite orientations ("in" and "out" orientations as defined by the authors). However, the "out" conformation is visualized within the complex of GephFL dimer and DARPin-27F3. In the tetramerized GephFL structure without DARPin-27F3, the linker adopts the "in" conformation. Does the binding of DARPin-27F3 change the orientation of the linker? Does the binding of DARPin-27F3 change the oligomerization state of GephFL?
- (4) The authors mentioned that the subdomain II mediates the assembly between two GephFL dimers via electrostatic interactions. Since the charged surface of subdomain II has been proposed to be involved in the binding and phase separation between GephE and GlyR/GABAAR, the formation of dimer-of-dimer interface might influence the binding of receptors. Besides, the C-terminal of the linker region adopts the "in" conformation and occupies the binding site of receptors. The interaction between Geph and receptors may also be affected. The authors need to discuss the potential impact of GephFL oligomerization state on receptor binding at the very least. Some additional experimental data would enhance the scientific value of the paper further.

Minor points:

- (1) The author mentioned that previous studies based on the P2 splicing variant of GephFL showed trimerized assembly and P1 GephFL used in this study adopts a dimer-based assembly mode. Could the authors discuss, based on the new results in this study, how can the splicing influence the oligomerization state of GephFL?
- (2) The authors generated a series of mutations to disrupt the dimer-of-dimer interface and validated these mutants by cell-based assays. Did the authors observe changes in fractionation profiles by gel filtration chromatography?

- (3) In Fig7D, it seems that there are multiple cells clustered together in the image of the “EEHE” group. Could the authors replace the image with a clearer view similar to other panels in the figure?
- (4) In line 189-195, the authors superimpose the structure of GephE-27F3 and GephE-GlyR and refer to FigS4. But the structure of FigS4 is from GephFL dimer-27F3 complex. Maybe this part of the analysis can be moved to the next section.
- (5) Panels d&e in Figure 5 appear to be swapped and do not match with the text and legend.
- (6) In line 130, the reference should be “Bai et al” rather than “Bei et al”.

Reviewer #2

(Remarks to the Author)

In this study, the authors investigate gephyrin oligomerization using a combination of cryo-EM, biochemical, and mutational approaches. Ortiz-López et al. establish that gephyrin assembles into stable dimers, which serve as fundamental building blocks for tetramers and potentially hexamers, in contrast to earlier trimer-based models. These findings provide important new insights into the structural logic underlying inhibitory postsynaptic scaffold organization. The experimental design is rigorous, and the purification of the P1 splice variant of gephyrin is particularly clean, enabling high-quality structural analysis. Notably, the authors succeed in resolving structural features of the linker region and identify two conformational states (“in” and “out”) with potential functional consequences for receptor binding at inhibitory synapses. Furthermore, they map phosphorylation sites that directly link structural conformation to regulatory control, offering a compelling mechanistic framework.

While the work makes an important contribution to our understanding of inhibitory synapse assembly, some points would benefit from clarification or further discussion. For example, the structural analyses are carried out on the P1 isoform expressed in *E. coli*, whereas previous SAXS studies using the P2 isoform supported a trimeric organization. How splice-variant-specific oligomerization translates to neuronal contexts *in vivo* remains unresolved, and additional discussion of this point would be useful. In addition, the GephG domain was not sufficiently visualized, even in full-length datasets, which limits the conclusions regarding N-terminal contributions to higher-order assembly. Finally, while the cellular assays provide evidence for the functional relevance of key mutations, additional validation in neurons would further strengthen the connection between the structural findings and postsynaptic architecture *in vivo*.

Specific suggestions:

- 1-It would be helpful if the authors could comment on whether insertion of cassette C4 in the P2 isoform may perturb linker conformation or its positioning relative to GephE, potentially explaining the distinct oligomeric behaviors observed between isoforms. Do they expect both isoforms to coexist or transition dynamically in neurons?
- 2-The N-terminal GephG domain is not visualized in the presented reconstructions. Can the authors comment on whether GephG may contribute to higher-order assembly and whether its flexibility is likely to be biologically relevant?
- 3-While 27F3 is a great tool for structure determination, it partially occludes one receptor-binding site on GephE. Could the authors address how confident they are that the observed conformations of GephE and the linker are not biased by DARPin binding?
- 4-The condensation assays were performed in HEK293T cells. Do the authors expect similar effects in neuronal contexts, where the protein environment and post-translational modifications are more complex?
- 5-The structural data highlight T324 and S325 as poised to regulate the “in/out” conformational switch. Have the authors attempted phospho-mimetic or phospho-null substitutions at these sites? If not, could they comment on the utility of such experiments to validate their models?
- 6-The evidence for tetrameric organization is strong, with atomic-level interfaces clearly resolved. By contrast, the proposed hexamer is supported only indirectly by SEC-MALS and low-resolution 2D classes. Could the authors clarify whether the hexamer represents a discrete higher-order state stabilized by specific contacts, or whether it instead reflects a more transient, end-to-end extension of the tetramer? Additional discussion of the molecular mechanism underlying possible hexamer formation would be valuable. In particular, it is difficult to visualize how a gephyrin dimer-of-dimers could accommodate interaction with a third dimer.

Reviewer #3

(Remarks to the Author)

I am writing to provide my comments on the submitted article titled “The manuscript is entitled “Cryo-EM Structures of Higher Order Gephyrin Oligomers Reveal Principles of Inhibitory Postsynaptic Scaffold Organization”. I appreciate the author’s efforts for isolating the protein and characterizing it with various orthogonal techniques to represent the oligomeric state of the protein.

The manuscript is well-organized, with clear articulation of objectives, methods, results, and conclusions. It addresses an important and timely research question, employing robust methodology and delivering significant contributions to its field. The clarity of presentation and rigor of analysis demonstrate scholarly depth and relevance to the journal’s audience. The manuscript has been reviewed for Isolation, extraction and Characterization of protein by AEX and SEC, characterizing through SEC-MALS, Blue Native PAGE, SDS-PAGE, DSC and Circular Dichroism and following points were observed

during detailed review.

1. The method for AEX and SEC and other logistic details should be mentioned in the supplement material provided along with other details for materials (Columns, Instrument, Software used).

2. X- axis Legend missing at the figure 1A for SEC-MALS, If HP-SEC MALS is used for molecular weight determination, please present the chromatograms representing the RT (min). Mention the MALS instrument setting used for the determination of Molecular weight.

Please present the SEC-MALS chromatograms of analyzed fractions with (molar mass g/mol) on Y-axis and X-axis volume (ml), so that true molecular mass can be verified from the provided mol mass in the manuscript.

3. SDS-PAGE a has been performed on some of fractions, please provide the method details either in the main manuscript or supplement file.

4. The author conducted the unfolding study utilizing the CD spectrometer, which highlights certain observations.

i. The samples should be scanned in the Far UV range (190 nm -250 nm) to represent the secondary structure of the protein, since the Gephyrin is present in the Helix or beta sheet form.

ii. After conforming the secondary structure, the unfolding study and temperature study should be performed to determining the melting temperature at which the protein changes its confirmation structure.

iii. Since the higher molecular species/oligomer are present the near UV spectra should be reported.

5. Figure presented at 4F (p.no 16) seems like the thermogram from DSC but in figure y axis legend represents the ellipticity value which is incorrect it should be Heat flow (mW.g-1) OR explain it.

Detailed instrument conditions are required for DSC like Ramp temperature and Time.

Version 1:

Reviewer comments:

Reviewer #1

(Remarks to the Author)

The authors have addressed the comments and requests raised by this reviewer by a set of new experiments and text revisions. Although more quantitative data would be better, the new data presented in the revised manuscript is overall satisfactory. The revised manuscript is with improved quality. The results presented in this paper will be valuable for the field to understand how gephyrin, via its different assembly modes, can scaffold the inhibitory synaptic signaling complexes. I support publication of the manuscript in NC with one minor suggestion for the authors to consider:

The Discussion part of the manuscript is rather lengthy and some of the points discussed repeat the results section. The authors may consider shortening the Discussion section to make the paper easier to read.

Reviewer #2

(Remarks to the Author)

The authors have adequately addressed all of the points raised in my previous review. I have no further comments at this time, and the manuscript has improved substantially.

Reviewer #3

(Remarks to the Author)

No Further comments, accept for publication

Response to the reviewer's comments to Ortiz-López et al.

First, we would like to take this opportunity to thank the reviewers for their time and appreciate all their constructive criticisms which undoubtedly helped to improve our manuscript. Second, we greatly appreciate the overall positive evaluation that our manuscript received from all three reviewers. Please find below our point by point responses (in blue) to each of the reviewer's comments.

Reviewer 1:

Major points:

(1) The use of DARPin-27F3 for visualizing GephE and the dimerized GephFL structure is interesting. Based on the structure, the authors discussed the influence of DARPin-27F3 binding on the GlyR β -GephE interaction. However, since the interaction between GlyR β 3-loop and GephE is quite strong, the potential competition between DARPin-27F3 and GlyR for Geph binding needs more detailed biochemical analysis. This is a good opportunity to do this, as such characterization will also add value for using DARPin-27F3 in iPSD visualization in neurons as well as in other applications.

Considering this suggestion, we decided to perform size exclusion chromatography to analyze the interaction between GephE, DARPIN 27F3 (referred to as 27F3) and a construct containing the intracellular loop (residues 208-341) of the GlyR β -subunit fused to coiled-coil repeats of the GCN4 transcription factor (referred to as GlyR- β GCN4), a similar construct as described in Ref. 3 (refers to references in this letter). As shown in Fig. 1 of this letter, we performed 4 different runs: GephE only, GlyR- β GCN4 only, the binary GephE+27F3 complex and a putative ternary GephE+27F3+GlyR- β GCN4 complex. GephE elutes at 14.6 ml, which, in the presence of 27F3 is shifted to 13.9 ml, thus demonstrating the expected complex formation. GlyR- β GCN4 alone elutes at 13.3 ml, a surprisingly early elution given a monomer mass of 21 kDa, even taking into account an oligomerization into dimers or trimers. The putative ternary GephE+27F3+GlyR- β GCN4 complex elutes as a peak at 13.6 ml with a leading shoulder at 12.9 ml. The leading shoulder is shifted to an earlier elution compared to the binary GlyR- β GCN4 complex, thus indicating formation of a ternary complex. This is confirmed by analysis of the corresponding fraction by SDS-PAGE (Fig. 1, inset), which demonstrates the presence of all three proteins. Due to the currently undefined oligomeric state of the GlyR- β GCN4 construct and the preliminary nature of these experiments they have not been incorporated into the manuscript.

Fig. 1: Size exclusion chromatography of GephE, the binary GlyR+ β GCN4 and GephE+27F3 complexes, and the ternary GephE+27F3+GlyR- β GCN4 complex. SEC was conducted on a Superdex 200 column at a salt concentration of 250 mM NaCl to avoid phase separation. The inset shows an SDS-PAGE analysis of the GephE+27F3+GlyR- β GCN4 complex together with the corresponding molecular mass of each protein (GephE 47 kDa, GlyR- β GCN4 21 kDa, DARPin 27F3 19 kDa).

In this context, we should also point out that a recent study by Macha et al. (Ref. 1) included a cryo-EM structure of a complex between GephE and a GlyR β -subunit derived peptide. In this structure only one GephE subunit was binding the peptide while the corresponding pocket in the other subunit was empty. Under these conditions binding of DARPin 27F3 would be straightforward resulting in a ternary complex as suggested by the SEC data shown in Fig. 1.

(2) According to the fractionation and structural analysis of the full-length Geph demonstrated in this paper, there exist multiple higher-order assemblies of GephFL mediated by the interaction between subdomain II from two GephFL dimers. But the trimerized G-domain could not be observed by Cryo-EM. Is the G-domain trimerization needed for the higher-order assembly of GephFL? Would GephE alone form larger oligomers at high concentrations?

After submission of our manuscript the aforementioned study by Macha et al. (Ref. 1) was published, which showed that the E-domain alone is sufficient for forming what they observed as filamentous structures. As in our structure of the dimer of dimers, this oligomerization is mediated via subdomain II in an analogous manner, thus demonstrating that the E-domain itself can mediate this interaction. We have referenced and briefly discussed this new structure on page 35 of the revised manuscript. Nevertheless, analysis of the SEC data from our GephFL purifications, specifically fraction A2' shown in Fig. 1e, demonstrates that the GephFL dimer of dimers is stable even at low concentrations, thus suggesting a stabilizing role for the linker region and/or GephG in GephE-mediated oligomerization.

Regarding a structural characterization of GephG we were unable to generate satisfactory 3D volumes from samples containing just GephFL dimers, presumably due to the small particle size and potentially preferred orientations. Nevertheless, upon further analysis of our data for the sample containing the GephFL dimer of dimers, we could derive a low-resolution 3D volume which previously escaped our attention. This volume fits well, both in size and shape, to a GephG trimer and a GephE dimer in which

the GephG trimer is located adjacent to subdomain II of GephE of one monomer with subdomain II of the second monomer being in an unbound state. As this reconstruction is based on only ~8500 particles, no high-resolution structure could be derived. Nevertheless, this volume places the GephG trimer in close proximity of subdomain II of GephE, thus suggesting that GephG could regulate the subdomain II-mediated oligomerization observed in the dimer of dimers. This low-resolution structure is presented in Fig. 5d, 5g and 5h in the revised manuscript and is described on page 19 and discussed on pages 34 and 35.

(3) The Cryo-EM analysis of GephFL dimer and tetramer showed two distinct conformations of the C-terminal linker regions with opposite orientations (“in” and “out” orientations as defined by the authors). However, the “out” conformation is visualized within the complex of GephFL dimer and DARPin-27F3. In the tetramerized GephFL structure without DARPin-27F3, the linker adopts the “in” conformation. Does the binding of DARPin-27F3 change the orientation of the linker? Does the binding of DARPin-27F3 change the oligomerization state of GephFL?

Regarding potential changes in the linker orientation, several lines of evidence support the existence of a discrete state of the linker stabilizing GephE in the absence of 27F3, not only experiments described in this manuscript but also published data as summarized below:

- 1) Our differential scanning calorimetry and circular dichroism experiments comparing both GephE₃₁₈ (residues 318-736) and GephE₃₀₉ (residues 309-736) showed an increase of 2.4-3.6°C in thermal stability, which we attribute to the new linker-subdomain II interface we structurally characterized. In addition, our micro-array experiments clearly demonstrate that the region ³¹⁴RRHR³¹⁷ (residues 314-317) is a hotspot of interaction with GephE. Together these data provide experimental proof for the significance of this molecular interface.
- 2) Using molecular dynamics simulations Liu et al. (Ref. 2) suggested that the same region of the linker (and the neighboring residues) interact with subdomain II of GephE, even though these MD simulations did not yield the same level of resolution.
- 3) Bai et al. (Ref. 3) demonstrated that electrostatic interactions between the strongly negatively charged subdomain II and the positively charged linker region are critical for phase separation.
- 4) The AlphaFold model (entry AF-Q9NQX3-2-F1-v6) for the full-length monomeric gephyrin shows the same conformation for the relevant residues in the linker. Based on this prediction we generated a model for the dimer (Fig. 2 of the rebuttal letter). The prediction is in excellent agreement with our cryo-EM structure with the extra residues adopting the out conformation.

All these observations were derived in the absence of 27F3, so we conclude that the “out” conformation of the linker is not induced by DARPin binding.

Fig. 2: AlphaFold prediction (AF-Q9NQX3-2-F1-v6) of N-terminally extended GephE. Residues 305-318 adopting the out-conformation are shown in red. The AlphaFold prediction was for the monomeric state of the full-length protein. To generate the GephE dimer the AlphaFold model was superimposed onto each monomer of a dimeric GephE crystal structure (PDB entry 4PD0, and the additional residues N-terminal to position 305 were truncated.

To analyse potential changes in the oligomerization state induced by 27F3, we performed size exclusion chromatography to study binding of this DARPin to the dimer of dimers (Figure 3 of the rebuttal letter). We observed that 27F3 does not dissociate the dimer of dimers into individual dimers, as no shift towards the dimeric fraction was observed. In fact, the peak of the complex (blue) elutes at 13.1 ml, slightly earlier than the dimer of dimers at 13.2 ml as expected due the presence of the bound DARPin. At the same time, the GephFL dimer elutes significantly later (15.4 ml).

Fig. 3: SEC analyses of the GephFL dimer, the GephFL dimer of dimers and the GephFL dimer of dimers in complex with 27F3. SEC was performed on a Superose 6 column. The peak at an elution volume of 19.5 ml corresponds to free 27F3.

(4) The authors mentioned that the subdomain II mediates the assembly between two GephFL dimers via electrostatic interactions. Since the charged surface of subdomain II has been proposed to be involved in the binding and phase separation between GephE and GlyR/GABA_AR, the formation of dimer of dimer interface might influence the binding of receptors. Besides, the C-terminal of the linker region adopts the “in”

conformation and occupies the binding site of receptors. The interaction between Geph and receptors may also be affected. The authors need to discuss the potential impact of GephFL oligomerization state on receptor binding at the very least. Some additional experimental data would enhance the scientific value of the paper further.

The reviewer raises an important point. As mentioned already in the original manuscript (now present on page 27 of the revised manuscript), we hypothesize that the “in” conformation is modulating the receptor binding as the interactions involved in that conformation (π -type interactions) are blocking the access to the receptor, but they are weaker than those involved in the receptor-gephyrin interactions. Hence, we expect the residues from the linker to be easily displaced upon receptor binding. We have confirmed this hypothesis by performing size exclusion chromatography of the GephFL dimer of dimers and the Gly- β GCN4 construct. As shown in Fig. 4 of the rebuttal letter, binding of GlyR- β GCN4 induces a left-shift in the peak (at 13.5 ml) in comparison to the GephFL dimer of dimers alone (at 14.0 ml), thus documenting that the receptor can bind to the dimer of dimers.

Fig. 4: Size exclusion chromatography of the GephFL dimer of dimers in the absence (black) and presence of GlyR- β GCN4 (blue). The peak at 18 ml represents free GlyR- β GCN4 as this protein is present in a stoichiometric excess. SEC was performed on a Superose 6 column at a salt concentration of 250 mM NaCl.

Minor points:

(1) The author mentioned that previous studies based on the P2 splicing variant of GephFL showed trimerized assembly and P1 GephFL used in this study adopts a dimer-based assembly mode. Could the authors discuss, based on the new results in this study, how can the splicing influence the oligomerization state of GephFL?

As demonstrated in this study, there are interactions between GephE and residues from the C-terminal region of the linker. As shown in Fig. 5 of the rebuttal letter, splice cassette C4c is close to the interacting region between the linker and GephE. Therefore, it seems plausible that the presence of the C4c cassette could alter the orientation of the linker and the way it interacts with the GephE dimer of dimers (“in” conformation) and dimers (“out” conformation). We have added a sentence pointing out this possibility on page 31 of the revised manuscript. It is conceivable that other splice cassettes could also modulate the oligomerization state of GephFL, however, in

the absence of structural data, one can only speculate about this possibility and hence we did not discuss this further in the manuscript.

Fig. 5: Location of the C4c splice cassette with respect to the linker residues (305-317, underlined) which could be visualized in the structure.

Created in Biorender: Ortiz-Lopez, D. <https://BioRender.com/7wdnjjr>

(2) The authors generated a series of mutations to disrupt the dimer-of-dimer interface and validated these mutants by cell-based assays. Did the authors observe changes in fractionation profiles by gel filtration chromatography?

We have not yet tested the behaviour of the mutants in terms of oligomerization by size exclusion. Nevertheless, according to the structures, we expect that the 379 and 422 mutants should weaken or even disrupt the interaction of the GephFL dimer of dimers mediated by subdomain II, as they are essential for the electrostatic interactions that keep the complex stable. In this context we would like to point out that, in response to reviewer 2, we studied these variants also in hippocampal neurons. These experiments confirm the data we obtained with these variants in HEK293T cells.

(3) In Fig7D, it seems that there are multiple cells clustered together in the image of the “EEHE” group. Could the authors replace the image with a clearer view like other panels in the figure?

Fig. 7D has been revised according to this suggestion.

(4) In line 189-195, the authors superimpose the structure of GephE-27F3 and GephE-GlyR and refer to Fig. S4. But the structure of Fig. S4 is from GephFL dimer-27F3 complex. Maybe this part of the analysis can be moved to the next section.

In fact, the order that the reviewer suggest is more logical and the manuscript has been revised as suggested. The corresponding paragraph has been moved to section “Cryo-EM structure of the GephFL-27F3 complex”. Figures Fig. S4 and Fig. S5 have been switched accordingly.

(5) Panels d&e in Figure 5 appear to be swapped and do not match with the text and legend.

Thanks for pointing out this mistake. The text and legend have been corrected.

(6) In line 130, the reference should be “Bai et al” rather than “Bei et al”.

Thanks for spotting this typo which has been corrected.

Reviewer 2:

In this study, the authors investigate gephyrin oligomerization using a combination of cryo-EM, biochemical, and mutational approaches. Ortiz-López et al. establish that gephyrin assembles into stable dimers, which serve as fundamental building blocks for tetramers and potentially hexamers, in contrast to earlier trimer-based models. These findings provide important new insights into the structural logic underlying inhibitory postsynaptic scaffold organization. The experimental design is rigorous, and the purification of the P1 splice variant of gephyrin is particularly clean, enabling high-quality structural analysis. Notably, the authors succeed in resolving structural features of the linker region and identify two conformational states (“in” and “out”) with potential functional consequences for receptor binding at inhibitory synapses. Furthermore, they map phosphorylation sites that directly link structural conformation to regulatory control, offering a compelling mechanistic framework.

While the work makes an important contribution to our understanding of inhibitory synapse assembly, some points would benefit from clarification or further discussion. For example, the structural analyses are carried out on the P1 isoform expressed in *E. coli*, whereas previous SAXS studies using the P2 isoform supported a trimeric organization. How splice-variant-specific oligomerization translates to neuronal contexts *in vivo* remains unresolved, and additional discussion of this point would be useful. In addition, the GephG domain was not sufficiently visualized, even in full-length datasets, which limits the conclusions regarding N-terminal contributions to higher-order assembly. Finally, while the cellular assays provide evidence for the functional relevance of key mutations, additional validation in neurons would further strengthen the connection between the structural findings and postsynaptic architecture *in vivo*.

Specific suggestions:

1-It would be helpful if the authors could comment on whether insertion of cassette C4 in the P2 isoform may perturb linker conformation or its positioning relative to GephE, potentially explaining the distinct oligomeric behaviors observed between isoforms. Do they expect both isoforms to coexist or transition dynamically in neurons?

As already stated in response to reviewer 1, we expect that insertion of the C4c cassette in the P2 variant is affecting the positioning of the linker relative to the GephE domain. As can be seen in Fig. 5 of this letter, the C4c splice cassette is close to the region ³¹⁴RRHR³¹⁷ so the insertion of this cassette between residues 289 and 290 could modulate the interaction of residues in the linker with subdomain II of the other monomer. Specifically, the absence of positively charged residues in the extra 14

residues of P2 may interfere with the charge-charge interactions observed in our structure.

Regarding the second point, a transcriptomic analysis by dos Reis et al. (Ref. 4) demonstrated that gephyrin undergoes extensive alternative splicing, generating a heterogeneous repertoire of isoforms that coexist in neurons. The expression ratios of specific isoforms are dynamically regulated in an activity-dependent manner and vary during development. The heterogeneity of the large number of splice variants will be further enhanced by posttranslational modifications, and we expect that different conformations of gephyrin can easily interconvert and hence are present in neurons.

2-The N-terminal GephG domain is not visualized in the presented reconstructions. Can the authors comment on whether GephG may contribute to higher-order assembly and whether its flexibility is likely to be biologically relevant?

As mentioned in response to reviewer 1 (2nd major point), we have obtained for the first time a volume containing a GephG trimer and a GephE dimer (see modified Fig. 5 in the manuscript and additions in the text on page 18 in the Results section and pages 33 and 34 in the Discussion). This low-resolution structure immediately suggests that GephG can control higher-order assembly, however, as stated in response to reviewer 1, these data are preliminary in nature and require additional cryo-EM data collection to obtain an atomic model and subsequent functional studies to probe the importance of selected residues for this interaction.

Regarding the biological relevance of GephG oligomerization, we would like to add that Bedet et al. (Ref. 5) demonstrated that insertion of the C5 cassette disrupts trimerization of GephG by inserting 13 residues in between residues 98 and 99 in GephG. This variant shows a significant reduction in cluster size, but not in receptor binding affinity, which implies that GephG oligomerization is playing a role in clustering *in vivo*.

In addition, our Blue Native PAGE analysis documents that sample A2 is containing fractions with molecular weights between 242 kDa and 480 kDa, which fit well with the expected molecular weights for trimers (~250 kDa) and the dimer of dimers (~340 kDa), suggesting potential transitions between both species. This observation has been now added on page 7 of the revised manuscript.

3-While 27F3 is a great tool for structure determination, it partially occludes one receptor-binding site on GephE. Could the authors address how confident they are that the observed conformations of GephE and the linker are not biased by DARPin binding?

This issue was also raised by reviewer 1 (point 3). As stated before, several observations described in the literature agree with our structure including molecular dynamics simulations (Ref. 2) and electrostatic potential analysis (Ref. 3). Furthermore, our peptide array assays, circular dichroism spectroscopy and differential scanning calorimetry measurements conducted in the absence of the DARPin demonstrate a stabilizing effect of the linker on GephE, which will depend on the

interaction between subdomain II and the linker. In addition, an AlphaFold model of GephFL (shown in Figure 2 of this letter) predict the C-terminal part of the linker to be in the “out” conformation, in direct contact with subdomain II of the other monomer. These results cannot exclude that the linker could adopt additional conformations but support the existence of the “out” conformation as a discrete and observable conformational state.

4-The condensation assays were performed in HEK293T cells. Do the authors expect similar effects in neuronal contexts, where the protein environment and post-translational modifications are more complex?

In collaboration with the lab of Christian Specht (INSERM, Paris, France) we have now analyzed our mutants in hippocampal neurons after lentiviral transfection. These experiments demonstrate that all variants can still be recruited to inhibitory synapses, thus suggesting that disturbing key electrostatic interactions does not globally impair the ability of gephyrin to associate with GABA receptor-containing postsynaptic sites. Furthermore, we analysed the enrichment of gephyrin variants at synaptic densities by quantifying the mScarlet-gephyrin intensity. We found a similar trend than the one observed in HEK cells, where the Geph^{R379E} and Geph^{EEHE} variants were more affected than Geph^{D422N}. These experiments demonstrate that the electrostatic interactions stabilizing the dimer-of-dimers influence gephyrin organization at postsynaptic sites. These data have been added to the manuscript as Fig. 8.

5-The structural data highlight T324 and S325 as poised to regulate the “in/out” conformational switch. Have the authors attempted phospho-mimetic or phospho-null substitutions at these sites? If not, could they comment on the utility of such experiments to validate their models?

These experiments are planned as part of a future manuscript. In the revised version we decided to instead include studies of our interface mutants in neurons to complement the data obtained with these variants in HEK cells (see previous point).

We would like to point out that a recent study by Burdina et al. (Ref. 6) suggested that S325 phosphorylation is abolishing the interaction between collybistin and gephyrin in HEK cells. In the presence of collybistin, the cluster size of gephyrin was observed to be reduced, while overexpression of the S325D phosphomimic variant could restore cluster size to wildtype levels. These results could indicate that the “in” conformation observed in our structure could also be involved in the recognition of collybistin, and potentially other binders, as well as in regulating cluster size.

6-The evidence for tetrameric organization is strong, with atomic-level interfaces clearly resolved. By contrast, the proposed hexamer is supported only indirectly by SEC-MALS and low-resolution 2D classes. Could the authors clarify whether the hexamer represents a discrete higher-order state stabilized by specific contacts, or whether it instead reflects a more transient, end-to-end extension of the tetramer? Additional discussion of the molecular mechanism underlying possible hexamer

formation would be valuable. In particular, it is difficult to visualize how a gephyrin dimer-of-dimers could accommodate interaction with a third dimer.

As mentioned by the reviewer, the lack of a high-resolution structure is a limitation to elucidate the mechanism behind GephFL hexamerization. Nevertheless, as pointed out in our manuscript, significant conformational changes were observed between GephFL dimers and GephFL dimer of dimers, especially in subdomain IV (Fig. 9), which appears to be sensitive to changes in oligomerization state. Our hypothesis is that more conformational changes in that region could trigger hexamerization and in consequence, the formation of high order oligomers like the fibril-like structures observed by Macha et al. described in Ref. 1.

Reviewer #3:

I am writing to provide my comments on the submitted article titled "Cryo-EM Structures of Higher Order Gephyrin Oligomers Reveal Principles of Inhibitory Postsynaptic Scaffold Organization". I appreciate the author's efforts for isolating the protein and characterizing it with various orthogonal techniques to represent the oligomeric state of the protein.

The manuscript is well-organized, with clear articulation of objectives, methods, results, and conclusions. It addresses an important and timely research question, employing robust methodology and delivering significant contributions to its field. The clarity of presentation and rigor of analysis demonstrate scholarly depth and relevance to the journal's audience. The manuscript has been reviewed for Isolation, extraction and Characterization of protein by AEX and SEC, characterizing through SEC-MALS, Blue Native PAGE, SDS-PAGE, DSC and Circular Dichroism and following points were observed during detailed review.

1. The method for AEX and SEC and other logistic details should be mentioned in the supplement material provided along with other details for materials (Columns, Instrument, Software used).

Methods details regarding both AEX and SEC chromatography have been now added to the Methods section under the heading "Plasmid constructs, protein expression and purification".

2. X- axis Legend missing at the figure 1A for SEC-MALS, If HP-SEC MALS is used for molecular weight determination, please present the chromatograms representing the RT (min). Mention the MALS instrument setting used for the determination of Molecular weight.

Please present the SEC-MALS chromatograms of analyzed fractions with (molar mass g/mol) on Y-axis and X-axis volume (ml), so that true molecular mass can be verified from the provided mol mass in the manuscript.

We thank the reviewer for this comment, as in the way it was written our procedure was not clear. The protein concentration was determined using the refractive index

detector (Optilab T-rEX) while the UV signal was not used to measure the concentration. This point has now been clarified in the Methods section. In addition, we provided additional experimental details on pages 38-39.

The x-axis legend in Fig. 1A has been added accordingly. In Fig. 1g, 1h and 1i (SEC MALS data) the label for the y-axis (molar mass in kDA) was presented already on the right side of each panel.

3. SDS-PAGE a has been performed on some of fractions, please provide the method details either in the main manuscript or supplement file.

SDS-PAGE details have now been added to the Methods section, under the title “SDS-PAGE and Blue Native PAGE”. We thank the reviewer for this comment.

4. The author conducted the unfolding study utilizing the CD spectrometer, which highlights certain observations.

i. The samples should be scanned in the Far UV range (190 nm -250 nm) to represent the secondary structure of the protein, since the Gephyrin is present in the Helix or beta sheet form.

Thanks for mentioning this point. In fact, the far UV CD spectra were measured before performing the unfolding experiments to confirm no significant changes in secondary structure. These data have been added to the supplement as Fig. S4a.

ii. After conforming the secondary structure, the unfolding study and temperature study should be performed to determining the melting temperature at which the protein changes its confirmation structure.

Please see previous point.

iii. Since the higher molecular species/oligomer are present the near UV spectra should be reported.

We decided not to conduct these experiments for the following reason: Previous SEC analyses from both proteins (now added as Fig. S4b) indicated small leading shoulders in both GephE₃₀₉ and GephE₃₁₈, which we attribute to the formation of higher oligomers (presumably dimers of dimers). Due to the presence of dimers and dimer of dimers the respective CD signal would have been a mixture of the two states and hence the interpretation of the data would have been ambiguous.

5. Figure presented at 4F (p.no 16) seems like the thermogram from DSC but in figure y axis legend represents the ellipticity value which is incorrect it should be Heat flow (mW.g-1) OR explain it.

Thanks for spotting this mistake. The experiments were indeed mislabelled and have been corrected: d) corresponds to the DSC, e) to the peptide binding array and f) to the circular dichroism experiments.

Detailed instrument conditions are required for DSC like Ramp temperature and Time. Instrument conditions have been now added to materials and methods section, subsection “Differential Scanning Calorimetry (DSC)”.

References:

1. Macha A, Liebsch F, Bruckisch EHW, Burdina N, von Stülpnagel I, Benting K, Gunkel M, Behrmann E, Schwarz G. Gephyrin filaments represent the molecular basis of inhibitory postsynaptic densities. *Nat Commun.* 2025 Sep 16;16(1):8293. doi: 10.1038/s41467-025-63748-w.
2. Liu YT, Tao CL, Zhang X, Xia W, Shi DQ, Qi L, Xu C, Sun R, Li XW, Lau PM, Zhou ZH, Bi GQ. Mesophasic organization of GABA_A receptors in hippocampal inhibitory synapses. *Nat Neurosci.* 2020 Dec;23(12):1589-1596. doi: 10.1038/s41593-020-00729-w.
3. Bai G, Wang Y, Zhang M. Gephyrin-mediated formation of inhibitory postsynaptic density sheet via phase separation. *Cell Res.* 2021 Mar;31(3):312-325. doi: 10.1038/s41422-020-00433-1.
4. Dos Reis R, Kornobis E, Pereira A, Tores F, Carrasco J, Gautier C, Jahannault-Talignani C, Nitschké P, Muchardt C, Schlosser A, Maric HM, Ango F, Allemand E. Complex regulation of Gephyrin splicing is a determinant of inhibitory postsynaptic diversity. *Nat Commun.* 2022 Jun 18;13(1):3507. doi: 10.1038/s41467-022-31264-w.
5. Bedet C, Bruusgaard JC, Vergo S, Groth-Pedersen L, Eimer S, Triller A, Vannier C. Regulation of gephyrin assembly and glycine receptor synaptic stability. *J Biol Chem.* 2006 Oct 6;281(40):30046-56. doi: 10.1074/jbc.M602155200.
6. Burdina N, Liebsch F, Macha A, Gil JLO, Frommelt P, Rais I, Basler F, Pöpsel S, Schwarz G. Phosphoinositide- and Collybistin-Dependent Synaptic Clustering of Gephyrin. *J Neurochem.* 2025 Aug;169(8):e70169. doi: 10.1111/jnc.70169.

Reviewer #1 (Remarks to the Author):

“The Discussion part of the manuscript is rather lengthy and some of the points discussed repeat the results section. The authors may consider shortening the Discussion section to make the paper easier to read.”

Our answer:

We thank the reviewer for this suggestion. We have shortened the Discussion by removing passages that repeated information already presented in the Results, including specific numerical values and residue descriptions that had been previously discussed.